# A peroxisomal ubiquitin ligase complex forms a retrotranslocation channel

Peiqiang Feng[1 ✉], Xudong Wu[1], Satchal K. Erramilli[2], Joao A. Paulo[3], Pawel Knejski[2], Steven P. Gygi[3], Anthony A. Kossiakoff[2,4] & Tom A. Rapoport[1 ✉]

Peroxisomes are ubiquitous organelles that house various metabolic reactions and are essential for human health[1–4]. Luminal peroxisomal proteins are imported from the cytosol by mobile receptors, which then recycle back to the cytosol by a poorly understood process[1–4]. Recycling requires receptor modification by a membrane-embedded ubiquitin ligase complex comprising three RING finger domain-containing proteins (Pex2, Pex10 and Pex12)[5,6]. Here we report a cryo-electron microscopy structure of the ligase complex, which together with biochemical and in vivo experiments reveals its function as a retrotranslocation channel for peroxisomal import receptors. Each subunit of the complex contributes five transmembrane segments that co-assemble into an open channel. The three ring finger domains form a cytosolic tower, with ring finger 2 (RF2) positioned above the channel pore. We propose that the N terminus of a recycling receptor is inserted from the peroxisomal lumen into the pore and monoubiquitylated by RF2 to enable extraction into the cytosol. If recycling is compromised, receptors are polyubiquitylated by the concerted action of RF10 and RF12 and degraded. This polyubiquitylation pathway also maintains the homeostasis of other peroxisomal import factors. Our results clarify a crucial step during peroxisomal protein import and reveal why mutations in the ligase complex cause human disease.

Peroxisomes are membrane-bounded organelles present in most eukaryotic cells. They are involved in essential cellular metabolism, notably, oxidation of fatty acids and destruction of reactive oxygen species. Most peroxisomal metabolic enzymes reside in the lumen, but are synthesized in the cytosol and imported into the organelle[1–4]. The importance of peroxisomes is highlighted by human genetic disorders in their biogenesis, such as Zellweger syndrome[4,7].

Most luminal peroxisomal proteins contain a C-terminal targeting signal (PTS1) that consists of the sequence Ser-Lys-Leu or variants of it[1–4]. The PTS1 signal is recognized in the cytosol by the receptor Pex5 (and its paralogue Pex9 in some fungi), which binds to a docking complex on peroxisomes and subsequently delivers the cargo into the lumen. Whether Pex5 accompanies cargo into the lumen[8] or integrates into the membrane to become part of a translocation channel[9] is controversial, but at least a segment seems to move across the membrane. To start a new import cycle, Pex5 must ultimately be returned to the cytosol. This recycling requires monoubiquitylation of Pex5 at a conserved Cys near the N terminus and extraction by a hexameric double-ring ATPase, which comprises alternating Pex1 and Pex6 subunits[10]. How the receptor would re-emerge in the cytosol to enable monoubiquitylation and retrotranslocation across the membrane is unknown.

Import of other luminal proteins relies on a PTS2 signal, an N-terminal sequence that is recognized by Pex7 and additional receptors that in fungi include Pex18, Pex20 and Pex21 (refs. [1–4,11,12]). These receptors are similar to Pex5, as they are also monoubiquitylated at a conserved N-terminal Cys and returned to the cytosol by the Pex1–Pex6 ATPase. When the normal recycling of Pex5 or the other receptors is blocked, for example, by inactivating the Pex1–Pex6 ATPase, the receptors are instead polyubiquitylated on Lys residues and subsequently degraded by the proteasome. This alternative pathway has been termed 'receptor accumulation and degradation in the absence of recycling (RADAR)'[12].

Both monoubiquitylation and polyubiquitylation of the receptors are catalysed by a conserved membrane-embedded ubiquitin ligase (E3) complex, consisting of Pex2, Pex10 and Pex12 (refs. [5,6]). In yeast, monoubiquitylation also requires the ubiquitin-conjugating (E2) enzyme Pex4 and its membrane-anchored activator Pex22; in higher organisms, these may be supplanted by more promiscuous E2 enzymes[13]. Polyubiquitylation requires the E2 enzyme Ubc4 or its homologues[14]. Pex2, Pex10 and Pex12 all have ring finger domains[6], a hallmark of many ubiquitin ligases. However, the involvement of three ring finger proteins is unique and conflicting data exist on their roles in monoubiquitylation and polyubiquitylation[15,16]. It is also unclear whether the ligase complex only catalyses ubiquitylation of the receptors or also facilitates their retrotranslocation, analogously to how ubiquitin ligases mediate the retrotranslocation of misfolded proteins from the endoplasmic reticulum (ER) into the cytosol in ER-associated protein degradation (ERAD)[17].

[1]Department of Cell Biology, Howard Hughes Medical Institute, Harvard Medical School, Boston, MA, USA. [2]Department of Biochemistry and Molecular Biology, University of Chicago, Chicago, IL, USA. [3]Department of Cell Biology, Harvard Medical School, Boston, MA, USA. [4]Institute for Biophysical Dynamics, University of Chicago, Chicago, IL, USA. ✉e-mail: peiqiang_feng@hms.harvard.edu; tom_rapoport@hms.harvard.edu

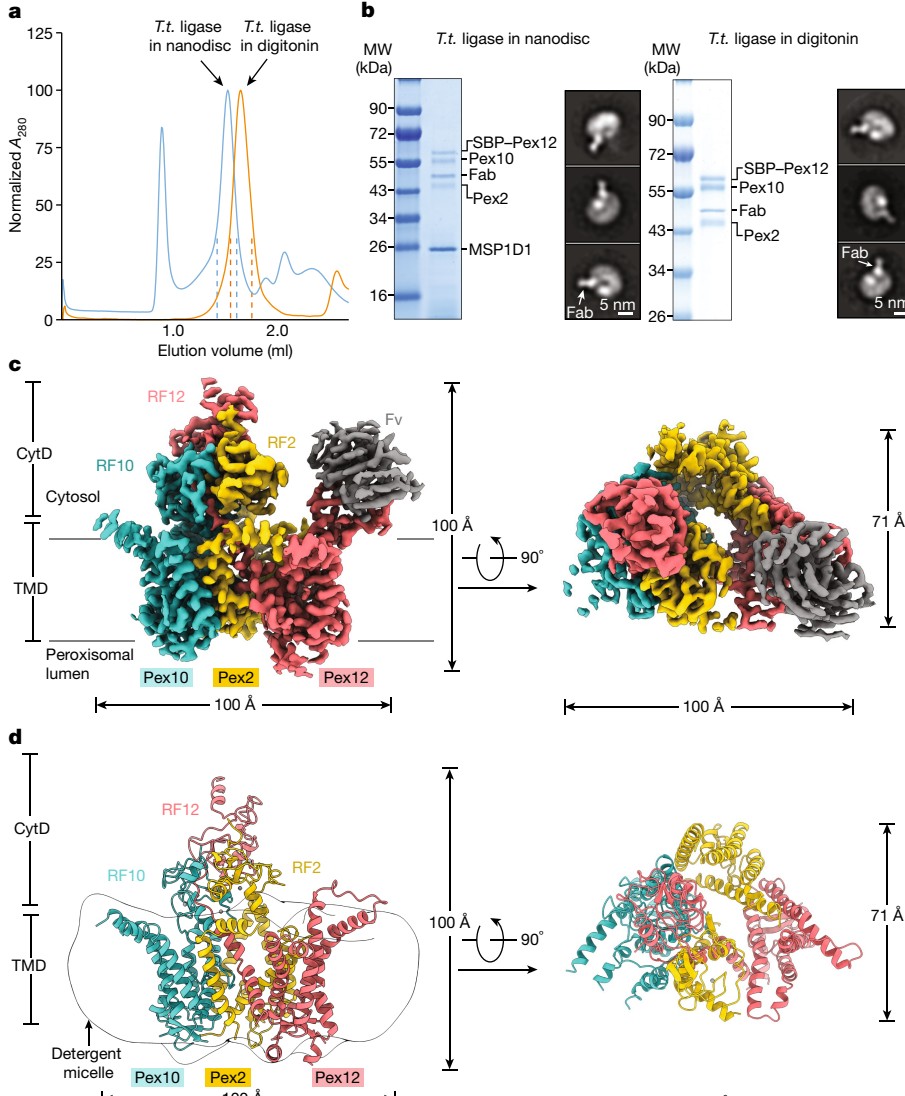

**Fig. 1 | Cryo-EM structure of the ligase complex. a**, Gel-filtration profile of the Fab-bound *T. thermophilus* (*T.t.*) ligase complex (Pex2 (140–495), Pex10 (1–454) and Pex12 (1–439) with an N-terminal streptavidin-binding peptide (SBP) tag) in nanodiscs or in digitonin. **b**, The peak fractions between the dashed lines in **a** were analysed by SDS–PAGE and Coomassie blue staining, as well as by negative-stain EM. MSP1D1 is the scaffold protein of the nanodiscs. The white arrow indicates the bound Fab. The results are representative of three biological repeats. For gel source data, see Supplementary Fig. 1. MW, molecular weight. **c**, Cryo-EM density map of the ligase complex with bound Fab, with views from the side and the cytosol. The map was sharpened with a *B* factor of −80.0 Å² and is shown contoured at a level of 0.041. CytD, cytosolic domain; Fv, variable domain of the Fab; TMD, membrane-embedded domain. **d**, Model of the ligase complex. The boundary of the detergent micelle is indicated.

## Structure determination

To elucidate the function of the peroxisomal ligase complex, we determined its cryo-electron microscopy (cryo-EM) structure, using Pex2, Pex10 and Pex12 from the thermophilic fungus *Thermothelomyces thermophilus*, which are highly homologous to the corresponding proteins in *Saccharomyces cerevisiae* and higher organisms (Extended Data Fig. 1). The three proteins were co-expressed in *Pichia pastoris* and purified in the detergent digitonin by affinity chromatography and gel filtration. They formed a stable 1:1:1 complex of approximately 150 kDa (Fig. 1a,b). A similar complex was obtained with the homologues from *S. cerevisiae* or *Chaetomium thermophilum* (Extended Data Fig. 2). To increase the size of the particles and facilitate orienting them for cryo-EM analysis, we used phage display mutagenesis to generate Fabs against the *T. thermophilus* ligase complex reconstituted into nanodiscs. One of the Fabs bound strongly to the ligase complex in both nanodiscs and digitonin (Fig. 1a,b), probably to a loop protruding from their surfaces (Fig. 1b). A cryo-EM structure of the Pex2–Pex10–Pex12–Fab complex in digitonin was determined at 3.1 Å overall resolution (Extended Data Fig. 3 and Extended Data Table 1). The density map (Fig. 1c) allowed model building for all parts of the proteins (Fig. 1d and Extended Data Fig. 3g), with the exception of some loops that were invisible and not conserved in other species (Extended Data Fig. 1). The Fab bound to a cavity formed by a cytosolic loop of Pex12 (Fig. 1c and Extended Data Fig. 3h).

## Architecture of the ligase complex

The three Pex proteins form a channel with their transmembrane segments and a cytosolic tower with their ring finger domains (Fig. 1c,d). Each protein has five transmembrane segments, most of which were missed in predictions because of their hydrophilicity[18]. The three proteins are not sequence related, but four of their five transmembrane segments form superimposable structures (Extended Data Fig. 4a),

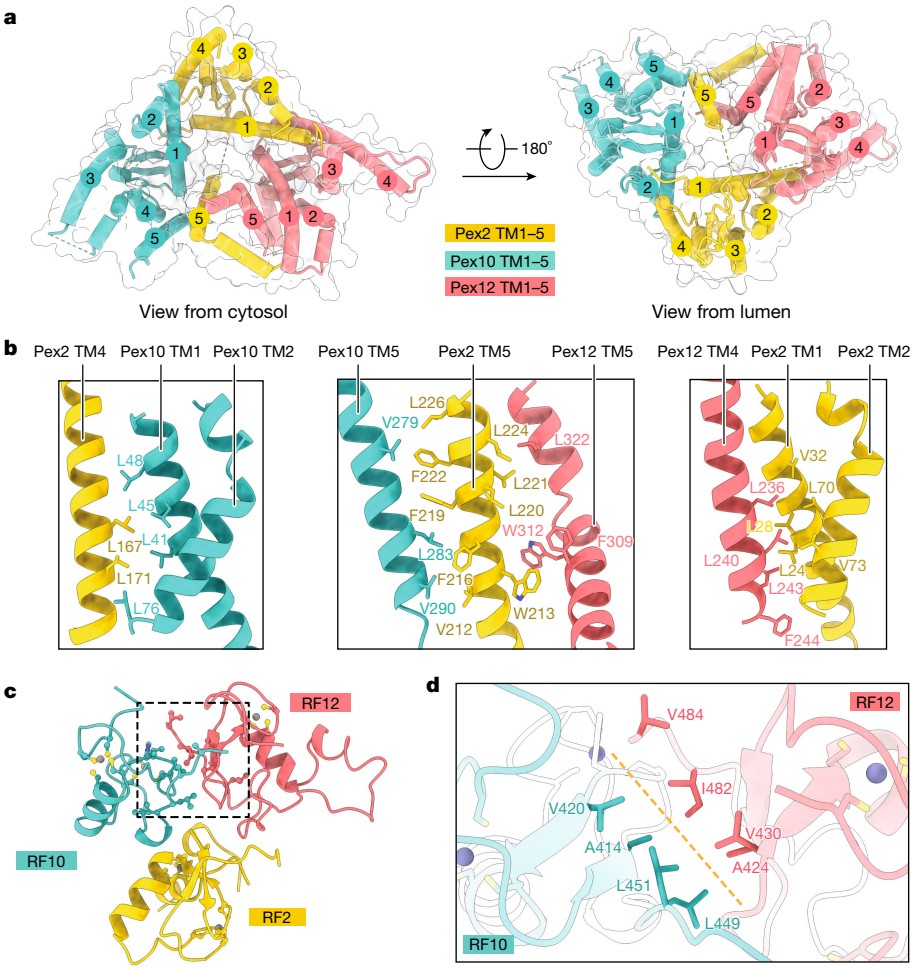

**Fig. 2 | Interactions between the Pex proteins. a**, Transmembrane segments of the ligase components, shown as cylinders. The transmembrane segments are numbered. The ring finger tower was omitted for clarity. The dashed lines indicate segments that were invisible in the density map. **b**, Interacting transmembrane segments are shown as cartoons. Hydrophobic amino acids at the interface are shown as sticks. **c**, Ring finger domains of the ligase complex are shown as cartoons. Cys residues are shown in yellow and Zn²⁺ atoms in grey. The dashed box shows the interface between RF10 and RF12, with interacting residues highlighted as sticks. **d**, Magnified view of the interface between RF10 and RF12 (orange dashed line), with interacting hydrophobic residues shown as sticks.

suggesting a common ancestor. The ring finger tower is formed by the membrane-proximal RF2 and RF10 domains and the more distal RF12 (Fig. 1d). RF12 is connected with the membrane-embedded domain of Pex12 through a central connector segment that passes through the interface between RF2 and RF10 and contains several conserved Pro residues (Extended Data Fig. 5a).

The membrane-embedded structure has a triangular shape (Fig. 2a). The five transmembrane segments of Pex10 and Pex12 form compact helical bundles, whereas the last transmembrane segment of Pex2 (transmembrane segment 5 (TM5)) is separated from TM1 to TM4. Pex2 associates with Pex10 and Pex12 extensively within the membrane, utilizing both TM5 and TM1–4 (Fig. 2b), whereas its ring finger interacts only weakly with the other ring finger domains (Fig. 2c) and does not bind to them in gel-filtration experiments performed with the isolated domains (Extended Data Fig. 5b,c). Conversely, Pex10 and Pex12 do not interact within the membrane (Fig. 2a) and are associated exclusively through their ring finger domains. RF10 and RF12 have an extensive interface (approximately 450 Å²) (Fig. 2c), mediated by several conserved hydrophobic residues (Fig. 2d and Extended Data Fig. 1), and the isolated domains interact in gel-filtration experiments (Extended Data Fig. 5d,e). The structure indicates that only the assembly of all three proteins can form a stable complex. Disease-causing point mutations map to the transmembrane segments, particularly to

clusters in Pex10 and Pex12, as well as to the three ring finger domains (Extended Data Fig. 6a,b).

## An open pore in the membrane

The transmembrane segments of the Pex proteins form a channel with an open pore (Fig. 3a,b). A short luminal Pex2 loop is next to the pore, but was invisible in the density map (indicated by a dashed line). Because this segment comprises only five small amino acids, it is unlikely to occlude the pore, leaving an opening of approximately 10 Å in diameter. Several residues surrounding the pore are hydrophilic and conserved (Fig. 3b and Extended Data Fig. 4b,c). A hydrophilic, luminal cavity of unknown significance is seen inside the membrane between the last transmembrane segments of Pex2 and Pex12 (Extended Data Fig. 4d). To test whether the open pore provides the translocation path for recycling peroxisomal import receptors and is therefore essential for overall protein import into peroxisomes, we introduced bulky residues into positions around the pore (Fig. 3c) and assessed import in *S. cerevisiae* cells. The protein import assay measures the conversion of tryptophan into a green pigment (prodeoxyviolacein (PDV)) by an engineered enzymatic cascade, in which the last enzyme (VioE) is sent into peroxisomes by an attached PTS1 signal and therefore can only participate in the cascade when its import is compromised[19]. Consistent with receptor

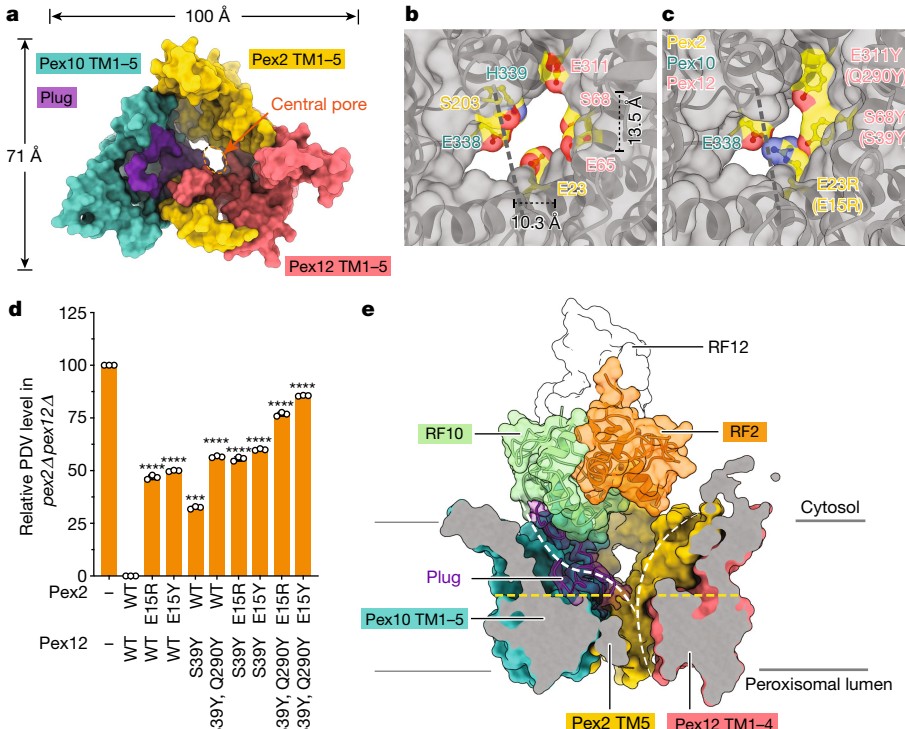

**Fig. 3 | Pore of the ligase complex. a**, Space-filling model of the membrane-embedded part of the ligase complex, viewed from the cytosol. The RF tower was omitted for clarity. The putative plug is shown in purple. **b**, Magnified view of the central pore, with residues lining the pore highlighted and viewed from the lumen. The dimensions of the pore are indicated. A short segment of a luminal loop, shown as a dashed line, is invisible in the density map. **c**, As in **b**, but with the predicted pore size reduced by the indicated mutations. The corresponding mutations in *S. cerevisiae* are given in brackets. **d**, Wild-type (WT) or mutant Pex2 or Pex12 were expressed as FLAG-tagged proteins from the endogenous promoter in *S. cerevisiae* cells lacking Pex2 and Pex12 (*pex2Δ pex12Δ*) (for expression levels, see Extended Data Fig. 5h). Peroxisomal protein import was determined by the reduction in the formation of a fluorescent pigment (PDV). Fluorescence was measured in cell lysates and the data were normalized, setting the fluorescence of *pex2Δ pex12Δ* cells as 100% and that of WT cells as 0%. The bar graphs show individual data points, the mean and the s.e.m. from the three biological repeats. Statistical significance between WT and mutants was calculated by one-way analysis of variance. ***$P < 0.01$ and ****$P < 0.0001$. See also Source Data file. **e**, Side view of a cut through a space-filling model of the membrane-embedded regions, with the plug shown as density with the embedded cartoon model. The ring finger domains are shown in full. The white dashed lines indicate two different routes for polypeptides from the lumen to the cytosol. The left path can only be taken when the plug is displaced. The yellow dashed line indicates the plane where pore-reducing residues are located (**c**,**d**).

translocation through the pore, protein import was progressively inhibited when the pore size was reduced by an increasing number of bulky residues (Fig. 3d and Extended Data Fig. 5g). Thus, recycling receptors probably access the catalytic ring fingers from the peroxisomal lumen, rather than sideways through the membrane, consistent with the absence of a lateral channel gate and with evidence that the receptor Pex5 enters the peroxisomal lumen before returning to the cytosol[8]. RF2 sits right above the pore, suggesting that a recycling receptor inserts its N-terminal segment into the pore, so that RF2 can catalyse monoubiquitylation at its conserved Cys. The other two ring fingers are more distant and cannot be reached by a polypeptide located inside the pore, as the path is blocked by a loop connecting the last transmembrane segment of Pex10 with its ring finger domain (Fig. 3a,e). However, this plug may be flexible and its sequence is poorly conserved (Extended Data Fig. 1). Indeed, the plug is not essential for protein import (Extended Data Fig. 5h), and in higher organisms, it seems to be shorter or absent.

The wall of the ligase channel consists of a single layer of transmembrane segments and contains small holes that are plugged by phospholipid and sterol molecules, with the hydrophilic head groups of phospholipids facing the channel interior (Extended Data Fig. 6c,d). The approximately 10 Å diameter of the pore may explain why the peroxisomal membrane is permeable to molecules smaller than approximately 800 Da[19,20]. The only other membranes permeable to small molecules are the outer membranes of mitochondria, chloroplasts and bacteria.

## Structure and function of the ring finger domains

The structure indicates that RF10 is a canonical ring finger, with two $Zn^{2+}$ atoms coordinated by Cys and His residues and two conserved loops predicted to interact with the E2–ubiquitin conjugate (E2–Ub)[21], one loop containing residue L288 and the other R324 (called the linchpin residue[22]) (Figs. 2c and 4a). By contrast, neither RF2 nor RF12 is a canonical ring finger. Although RF2 has two loops (L1 and L2) that are expected to interact with the E2 enzyme, these do not contain conserved amino acids seen in other ring fingers. RF2 also lacks the linchpin residue. RF12 deviates even more from canonical ring fingers. It contains only five, instead of seven, conserved Cys residues, and lacks the second $Zn^{2+}$ atom, as seen in both the cryo-EM structure (Fig. 2c) and a crystal structure of the *S. cerevisiae* homologue that we determined at 1.5 Å resolution (Extended Data Fig. 7a and Extended Data Table 2). RF12 is therefore not predicted to interact with an E2 enzyme.

The structural features of the ring finger domains are consistent with results of an in vitro polyubiquitylation assay that uses isolated ring fingers and the E2 enzyme Ubc4. RF10 has some low activity, whereas both RF2 and RF12 are inactive (Fig. 4b, lane 4 and Extended Data Fig. 8a). RF12 is an activator of RF10, as it greatly stimulated the polyubiquitylation by RF10 (Fig. 4b, lane 5), in agreement with previous results[5]. Mutations in RF10 that were predicted to abolish E2 binding (L288A or R324A) or the interaction with RF12 (L270A) had no or low activity (lanes 11, 13 and 7). The mechanism by which RF12 stimulates

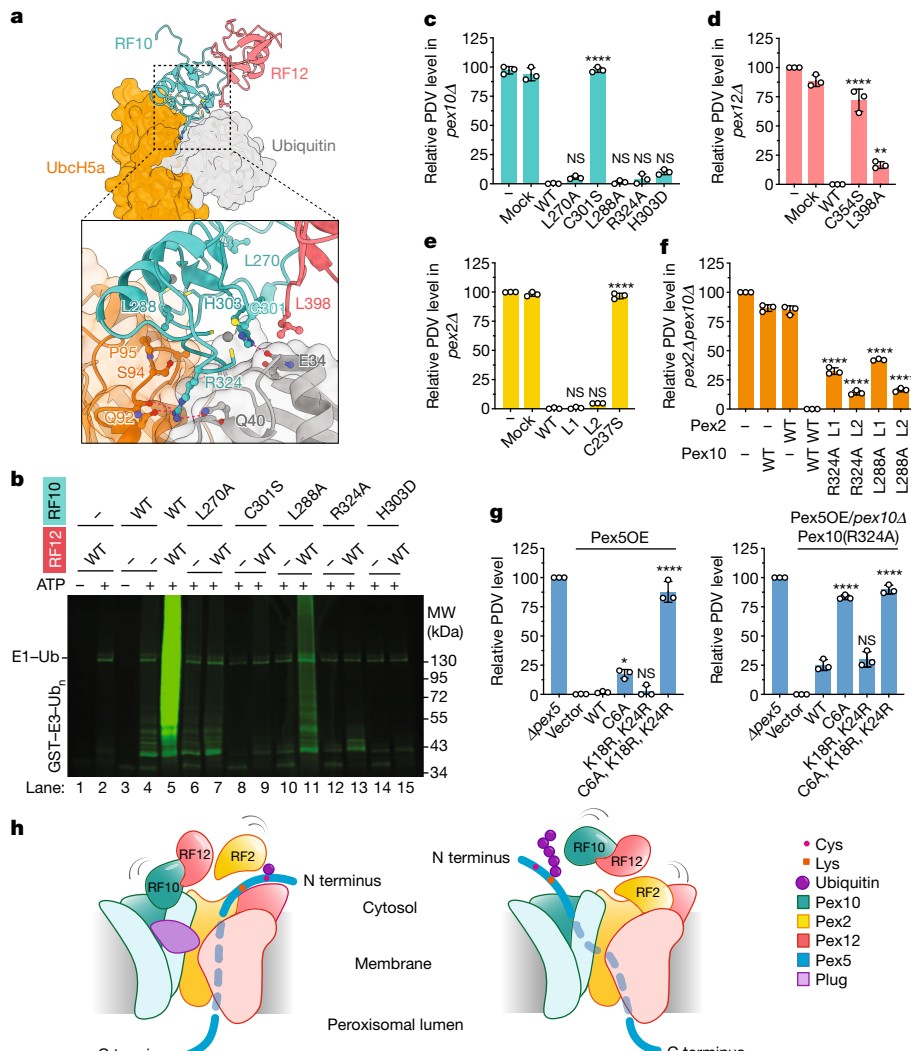

**Fig. 4 | Distinct functions of the ring finger domains of the ligase complex. a**, Predicted interaction of RF10 with E2–Ub. UbcH5a bound to ubiquitin (Protein Data Bank ID: 4AP4) was docked onto Pex10 of the RF10–RF12 complex. RF10 is a *S. cerevisiae* homology model derived from the *T. thermophilus* cryo-EM structure. RF12 is a crystal structure (Extended Data Fig. 7a). The magnified view shows amino acid interactions (red dashed lines). **b**, Polyubiquitylation reactions were performed with WT or mutant RF10 or RF12 in the presence of Uba1, Ubc4, Dylight800-labelled ubiquitin and ATP, as indicated. The results are representative of three biological repeats. For gel source data, see Supplementary Fig. 1. **c**, WT or mutant Pex10 was expressed in *S. cerevisiae* cells lacking Pex10 (*pex10Δ*). 'Mock' indicates cells expressing only an antibiotic resistance gene. Protein import was determined by the reduction

in fluorescent pigment (PDV) formation. Fluorescence in cell lysates was normalized, setting the fluorescence of *pex10Δ* cells as 100% and that of WT cells as 0%. Bar graphs show individual data points, the mean and the s.e.m. from three biological repeats. Statistical significance between WT and mutants were calculated by one-way analysis of variance. NS, not significant; *$P < 0.1$, **$P < 0.05$ and ****$P < 0.001$. See also Source Data file. **d**, As in **c**, but for *pex12Δ*. **e**, As in **c**, but for *pex2Δ*. **f**, As in **c**, but for *pex2Δ pex10Δ*. L1 and L2 indicate Pex2 loop mutations. **g**, WT or mutant Pex5 was overexpressed (Pex5OE) in WT cells or Pex10(R324A)-mutant cells. Vector, vector without Pex5. **h**, Receptor recycling pathway involving monoubiquitylation by RF2 (left model), and the degradation pathway involving polyubiquitylation by RF10 and RF12 (right model). RF10 is the active domain.

RF10 activity can be deduced from a comparison with homodimeric ring finger ligases, which are structurally most similar to the RF10–RF12 complex; alignment with structures of such ring finger dimers containing bound E2–Ub[23,24] predicts that L398 of Pex12 interacts with the end of the single helix in ubiquitin (Fig. 4a and Extended Data Fig. 7b,c). Mutation of L398 considerably reduced the stimulatory effect of RF12 in the in vitro assay (Extended Data Fig. 8b). Thus, RF10 and RF12 probably function as a unit in polyubiquitylation.

To test the role of RF10 and RF12 in peroxisomal protein import, we replaced endogenous Pex10 or Pex12 in *S. cerevisiae* with ring finger mutants defective in polyubiquitylation. The Pex10 mutants L270A, L288A, R324A and H303D, and the Pex12 mutant L398A, were as active as the corresponding wild-type proteins (Fig. 4c,d). Even the combination of Pex10(L288A) and Pex12(L398A) did not cause an import defect

(Extended Data Fig. 8c). Thus, polyubiquitylation by RF10–RF12 is dispensable for peroxisomal protein import and normal receptor recycling. Accordingly, human mutations (H310D/Q and R331Q), equivalent to *S. cerevisiae* H303D and R324Q, cause relatively mild disease phenotypes[25]. Conversely, mutations that are predicted to affect Zn[2+] binding and thus folding of the complex (C301S in Pex10 and C354S in Pex12) abolished import in vivo (Fig. 4c,d). Cys mutations in the ring fingers of the human proteins accordingly cause severe Zellweger syndrome[25].

To better understand RF10–RF12-dependent polyubiquitylation, we identified substrates of the pathway by quantitative proteomics in *S. cerevisiae*. In cells expressing the polyubiquitylation-defective Pex10 mutant R324A, the peroxisomal import receptors Pex5, Pex9 and Pex18 accumulated (Extended Data Fig. 9). Substrates also included the components of the receptor docking complex[1–4], particularly

Pex13 (Extended Data Fig. 9). Other peroxisomal proteins remained unchanged. These data suggest that the polyubiquitylation pathway not only mediates the degradation of receptors (RADAR pathway) but also maintains the homeostasis of other import factors, in agreement with the reported ligase-dependent degradation of several peroxisomal membrane proteins[26–28].

Next, we tested the role of the third ring finger (that is, RF2). An RF2 mutation designed to prevent $Zn^{2+}$ binding (C237S) abolished protein import (Fig. 4e), but did not affect the assembly of the ligase complex (Extended Data Fig. 5f). Mutations in the L1 and L2 loops of RF2 (P223A/R224D and D257A), which are predicted to reduce the interaction with the E2 enzyme, caused no import defects (Fig. 4e). However, when these mutations were combined with the polyubiquitylation-deficient Pex10 mutations R324A or L288A, which on their own were also benign, import was significantly reduced, particularly with the L1 mutations (Fig. 4f). Thus, receptor monoubiquitylation by RF2 was probably compromised in the loop mutants, but compensated for by Pex10Pex12-mediated polyubiquitylation.

To further test the role of RF2 in the monoubiquitylation pathway, we overexpressed Pex5 mutants in *S. cerevisiae* cells containing endogenous Pex5. Overexpression of Pex5(C6A), a mutant that lacks the Cys that is normally modified by monoubiquitin[14,29,30], caused only a minor import defect in wild-type cells (Fig. 4g). Thus, the C6A mutant did not block the ligase channel for endogenous Pex5, even though it cannot be extracted from the pore by the monoubiquitylation pathway. Instead, the pore seems to be cleared of the overexpressed mutant protein by the polyubiquitylation (that is, RADAR) pathway. Indeed, import was drastically reduced (Fig. 4g) when the C6A mutation was combined with Lys mutations that preclude Pex5 polyubiquitylation (K18R and K24R)[14,29,30]. The K18R/K24R mutations on their own caused no import defect (Fig. 4g). In agreement with RF10–RF12 mediating polyubiquitylation of Pex5, the C6A mutation caused similarly strong import defects when expressed in cells compromised in polyubiquitylation by an RF10 mutation (R324A) (Fig. 4g). By contrast, little effect was seen when the C6A mutant was expressed in a strain carrying mutations in the L1 loop of RF2 (Extended Data Fig. 8d). These results therefore suggest that RF2 mediates the normal monoubiquitylation in the receptor recycling pathway, whereas polyubiquitylation by RF10–RF12 prevents the clogging of the ligase channel when the recycling pathway is compromised. Consistent with RF2 catalysing monoubiquitylation, import was reduced by approximately 50% (compared to wild-type Pex5) when the Pex5 K18R/K24R mutant was expressed in the RF2-mutant strain (Extended Data Fig. 8d), whereas no import defect was observed in the RF10-mutant strain (Fig. 4g). It should be noted that we have not been able to demonstrate monoubiquitylation of Pex5 in vitro with the isolated RF2 or the full-length ligase complex, possibly because this reaction may require the proper insertion of the receptor into the ligase pore or the presence of another component.

## A model for ligase function

Our results support a model in which a peroxisomal import receptor enters the lumen of the organelle and is then exported back into the cytosol[8]. During export, the unstructured N-terminal segment would insert into the channel pore from the luminal side, such that the conserved Cys residue can be modified by the combined action of RF2 and Pex4 (or, in higher organisms, another E2) (Fig. 4h). The monoubiquitylated receptor would subsequently be extracted into the cytosol by the Pex1–Pex6 ATPase. When receptor recycling is compromised, the N-terminal segment would move laterally to reach RF10, a movement that requires plug displacement (Fig. 4h). RF10 would then cooperate with RF12 and Ubc4 (or its homologues) to polyubiquitylate the receptor on nearby Lys residues. The polyubiquitylated receptor would then be extracted from the ligase channel by another ATPase, presumably Cdc48 (p97 or VCP in mammals), and

finally degraded by the proteasome. This polyubiquitylation pathway is also used to degrade other peroxisomal membrane proteins. In our cryo-EM structure, the predicted interactions of RF2 and RF10 with E2 enzymes would lead to steric clashes (Extended Data Fig. 7d,e), indicating that both ring fingers need to undergo a conformational change to become active. Because RF2 and RF10 are connected by flexible loops with the membrane-embedded regions, these conformational changes would not affect the pore.

The overall pathways of peroxisomal receptor recycling and degradation resemble ERAD-L, in which misfolded luminal ER proteins are retrotranslocated by multispanning ubiquitin ligases, polyubiquitylated, and then extracted from the membrane by the Cdc48 ATPase[17,31]. However, in ERAD-L, the Hrd1 ubiquitin ligase and the associated Der1 protein each form a 'half-channel' with cytosolic and luminal cavities, respectively; translocation occurs through a thinned membrane region, rather than through an aqueous pore[32]. The peroxisomal retrotranslocon resembles more the classic Sec61/SecY translocon, as they both have aqueous pores through which polypeptides are translocated[33–35].

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

# Methods

## Cloning, plasmid construction and strains

All constructs were cloned using Gibson Assembly. For overexpression of the ligase complex in *P. pastoris*, DNA sequences coding for *T. thermophilus, S. cerevisiae* or *C. thermophilum* components were synthesized and codon optimized by GeneArt of Life Technology. Pex12 contained an N-terminal SBP tag followed by a HRV3C protease cleavage site, Pex10 contained a C-terminal haemagglutinin (HA) tag, and Pex2 contained a C-terminal FLAG tag. All genes were cloned between the EcoRI and NotI sites of the vector pPICZA-LINK, which was modified from pPICZA to allow the expression of multiple genes[36]. For expression of ring fingers in *Escherichia coli*, DNA sequences of *S. cerevisiae* genes were synthesized and cloned between the BamHI and NotI sites of the vector pGEX6P-1 containing an N-terminal glutathione *S*-transferase (GST) tag followed by a HRV3C protease cleavage site. Variants of all constructs were generated using Quikchange (Agilent).

The *S. cerevisiae* wild-type strain UTL7A (MATa, ura3-52, trp1, leu2-3/112) was kindly provided by R. Erdmann (Ruhr-University Bochum). All single and double knockout strains were constructed using standard transformation techniques and the pFA6A-NatMX or KanMX plasmids. The FLAG-tagged versions of Pex2, Pex10 and Pex12 were introduced at their endogenous locus by homologous recombination using the pFA6A-HygMX vector. For Pex5 overexpression, a HA tag followed by a 6×His tag was fused to the C terminus of full-length Pex5. The gene contained approximately 500 bp upstream of the ATG start codon, so that Pex5 was expressed from its endogenous promoter. The sequence was inserted into the multiple-cloning site of the CEN plasmid pRS416 (ref. [32]). All transformations of *S. cerevisiae* cells were done with the LiAc-PEG method[37].

## Protein expression, purification and nanodisc reconstitution

The *P. pastoris* wild-type strain SMD1168 was obtained from Life Technology. Transformations were performed by electroporation, following the manufacturer's instructions. Transformed yeast cells were grown on Zeocin-containing YPDS (YPD medium plus sorbitol) plates at 30 °C for 3 days. A single colony was picked to inoculate a starting culture, which was incubated at 30 °C overnight. A large culture was then inoculated by diluting the starter culture 1:100 into buffered minimal glycerol (BMG) medium. The culture was incubated at 30 °C for about 24 h and protein expression was induced by switching to the same volume of buffered minimal methanol (BMM) medium. After incubation for about another 20 h at 28 °C, the cells were harvested by centrifugation at 4,500*g* for 10 min. The pellet was stored at −80 °C until further use.

Cell pellets of about 100 g were resuspended in 100–120 ml buffer A (25 mM HEPES pH 7.4, and 400 mM NaCl) supplemented with 2 mM phenylmethane sulfonyl fluoride (PMSF) and 2 µM pepstatin A. The cells were lysed in a BioSpec Beadbeater for 40 min with 20 s/60 s on/off cycles in a water-ice bath. The homogenate was centrifuged at 10,000*g* for 20 min to remove cell debris. The supernatant was subjected to centrifugation in a Ti45 rotor (Beckman) at 44,000 r.p.m. for 1 h at 4 °C. The pelleted membranes were resuspended with a Dounce homogenizer in buffer A and pelleted again by centrifugation. The membranes were resuspended in 200–250 ml of buffer A containing 1% laurylmaltose neopentylglycol (LMNG) and a protease inhibitor cocktail and incubated for 60 min at 4 °C. Insoluble material was removed by centrifugation in a Beckman Ti45 rotor at 44,000 r.p.m. for 30 min. The supernatant was transferred to a new tube and incubated with 2 ml high-capacity streptavidin resin (Thermo Scientific) for 1 h. The resin was washed with 20–30 ml buffer A containing 0.1% digitonin (EMD Millipore), and bound protein was eluted with 10–15 ml of buffer B (25 mM HEPES pH 7.4, 150 mM NaCl, 10% glycerol and 0.1% digitonin) supplemented with 2 mM biotin (Sigma). The complex was concentrated with a 100-kDa cut-off Amicon filter (Sigma-Millipore) and further purified by size-exclusion chromatography on a Superose

6 3.2/300 Increase column (GE Healthcare), equilibrated with buffer C (25 mM HEPES pH 7.4, 150 mM NaCl and 0.05% digitonin).

For nanodisc reconstitution, a lipid stock was first prepared. The stock contained 10 mM each of 1,2-dioleoyl-sn-glycero-3-phosphocholine (DOPC), 1,2-dioleoyl-sn-glycero-3-phosphoethanolamine (DOPE) and 1,2-dioleoyl-sn-glycero-3-phospho-L-serine (DOPS) and was prepared as in ref. [38]. Protein purified in digitonin was incorporated into nanodisc using a 1:240:5 molar ratio of protein:lipid:membrane-scaffold protein 1D1 (MSP1D1). This mixture was incubated at 4 °C for 2 h with gentle agitation. Then, Bio-Beads (Bio-Rad) were added at 4 °C overnight with continuous rotation. The Bio-Beads were removed and the reconstitution mixture was centrifuged at 12,000*g* for 20 min to remove aggregated protein. The supernatant was loaded onto a Superose 6 3.2/300 Increase size-exclusion column (GE Healthcare) in gel-filtration buffer (25 mM HEPES pH 7.4 and 150 mM NaCl). Fractions containing nanodisc-reconstituted ligase complex were pooled and concentrated to about 10 µM. This material was used to generate Fabs.

To assemble a Fab–ligase complex, purified ligase complex in digitonin was incubated with Fab at a 1:1.5 molar ratio on ice for 1 h. The Fab–ligase complex was concentrated and loaded on a Superose 6 3.2/300 Increase size-exclusion column (GE Healthcare) in buffer C (25 mM HEPES pH 7.4, 150 mM NaCl, 0.05% digitonin). Peak fractions were pooled and concentrated to approximately 7 mg ml⁻¹ for cryo-EM analysis.

For the purification of GST-tagged ring finger domains, the proteins were expressed in the *E. coli* strain BL21(DE3). A cell lysate was first subjected to centrifugation at 10,000*g* for 30 min at 4 °C, and the filtered supernatant was applied to a glutathione resin (GE Healthcare). After elution with reduced glutathione, proteins were concentrated and loaded onto a Superdex 200 3.2/300 Increase size-exclusion column (GE Healthcare) equilibrated in gel-filtration buffer (25 mM HEPES pH 7.4, 150 mM NaCl, 1 mM Tris (2-carboxyethyl) phosphine hydrochloride (TCEP) and 5% glycerol). Peak fractions were pooled and stored at −80 °C until use.

For purification of the Pex12 ring finger domain used for crystallization, GST-3C–Pex12 RF bound to the glutathione resin was incubated with PreScission protease at 4 °C overnight to cleave off the GST tag. The flow-through fraction was collected and loaded onto a Mono Q ion-exchange column (GE Healthcare). The protein was eluted with a salt gradient (0–500 mM NaCl and 25 mM HEPES pH 7.4) and peak fractions were pooled. The fractions corresponding to the Pex12 ring finger domain were concentrated and applied to a HiLoad 16/60 Superdex 75 gel-filtration column (GE Healthcare) pre-equilibrated in gel-filtration buffer (25 mM HEPES, pH 7.4, 150 mM NaCl, 1 mM TCEP and 5% glycerol). The purified protein was concentrated to 6 mg ml⁻¹, flash-frozen in liquid nitrogen and stored at −80 °C.

## Crystallization of the *S. cerevisiae* Pex12 ring finger domain

The Pex12 ring finger domain was crystallized by the hanging-drop vapour diffusion method by mixing 0.2 µl of the protein at 6 mg ml⁻¹ with 0.2 µl of the 100 µl reservoir solution containing 0.1 M bis-Tris propane, pH 8.5, 0.2 M sodium fluoride and 20% PEG3350. Crystals were cryo-protected by gradually increasing the glycerol concentration in the drop by repeated additions of well solution supplemented with 30% glycerol. The crystal was removed from the drop and swiped through another drop of well solution supplemented with 30% glycerol and then flash-frozen in liquid nitrogen. Data were collected at 100 K at the 24-ID-C beamline at the Advanced Photon Source (APS). Data were indexed, integrated and scaled with HKL2000/3000 packages to 1.5 Å resolution.

## Identification of ligase complex-specific Fabs using phage display

The purified *T. thermophilus* ligase complex was reconstituted into nanodiscs as described above, except that chemically biotinylated

MSP1D1 was used. The efficiency of biotinylation was evaluated by capturing the nanodiscs with streptavidin-coated paramagnetic beads (Promega). For Fab selection, a previously described Fab phage library was used[39]. The nanodiscs containing the ligase complex and the library were diluted into selection buffer (20 mM HEPES pH 7.4, 150 mM NaCl and 1% BSA). Five rounds of sorting were performed as previously described[40,41]. In the first round, biopanning was performed manually using 400 nM of reconstituted ligase complex. To increase the stringency of selection pressure, four additional rounds of sorting were performed semi-automatically by stepwise reduction of the target concentration: 200 nM in the second round, 100 nM in the third round, 50 nM in the fourth round and 25 nM in the fifth round. All rounds of phage display were performed using a previously described protocol[39]. For each round except the first, the amplified phage population from each preceding round was used as the input pool. Of empty MSP1D1 nanodiscs, 2 μM was used throughout the selection as competitors.

## Single-point phage ELISA to validate Fab binding to the reconstituted ligase complex

Single-point phage ELISA was performed to validate unique binders obtained from the fourth and fifth rounds of phage display. Sequencing of individual colonies harbouring phagemids was performed at the University of Chicago Comprehensive Cancer Center DNA Sequencing facility, and unique clones were selected and phages were amplified before ELISA as previously described[40,41]. ELISA was performed using a previously described protocol[42].

## Fab cloning, expression, purification and validation

Specific binders based on phage ELISA results were sequenced at the University of Chicago Comprehensive Cancer Center DNA Sequencing facility and unique clones were then subcloned into the Fab expression vector RH2.2 (gift of S. Sidhu) using the In-Fusion Cloning kit (Takara). Successful cloning was verified by DNA sequencing. Fabs were then expressed and purified as previously described[35]. Following purification, Fab samples were verified for purity by 4–20% SDS–PAGE and subsequently dialysed overnight in 25 mM HEPES pH 7.4 and 150 mM NaCl. Purified Fab affinities were estimated by multi-point ELISA[42] using the ligase complex in biotinylated nanodiscs.

## Negative-stain EM

The ligase complex reconstituted in nanodiscs or in digitonin at a concentration of 0.02 mg ml[−1] was applied to glow-discharged continuous carbon grids (Electron Microscopy Sciences, Inc.). After 1 min of adsorption, the grids were blotted with filter paper to remove excess sample, immediately washed twice with 4 μl of 1.5% freshly made uranyl formate solution and incubated with 4 μl of 1.5% uranyl formate solution for an additional 30 s. The grids were then further blotted with filter paper to remove the uranyl formate solution, air dried at room temperature and examined with a Tecnai T12 electron microscope (Thermo Fisher Scientific) equipped with an LaB6 filament and operated at 120 kV acceleration voltage, using a nominal magnification of ×52,000 at a pixel size of 2.13 Å.

## Single-particle cryo-EM sample preparation and data acquisition

The concentrated sample was incubated with MS(PEG)12 methyl-PEG-NHS-ester (Thermo Fisher) at a 1:40 molar ratio for 2 h on ice to reduce the preferred orientation of particles on the grids[43]. Then, 3.0 μl PEGylated sample was applied to a glow-discharged quantifoil grid (1.2/1.3, 400 mesh). The grids were blotted for 7.0 s with a blot force of 12 at approximately 100% humidity and plunge-frozen in liquid ethane using a Vitrobot Mark IV instrument (Thermo Fisher Scientific).

Cryo-EM data were collected on a Titan Krios electron microscope (FEI) operated at 300 kV and equipped with a K2 Summit direct electron detector (Gatan) at the HHMI Janelia Farm Cryo-EM facility. An energy filter slit width of 20 eV was used to remove inelastically scattered electrons. All cryo-EM movies were recorded in super-resolution counting mode using SerialEM. The nominal magnification of ×81,000 corresponds to a calibrated physical pixel size of 1.06 Å and 0.53 Å in the super-resolution mode. The dose rate was 5.28 electrons Å[−2] s[−1]. The total exposure time was 10 s, resulting in a total dose of 52.8 electrons Å[−2] fractionated into 50 frames (200 ms per frame). The defocus range for the sample was between −1.0 and −2.5 μm.

## Data processing

A total of 9,019 dose-fractionated super-resolution movies were subjected to motion correction using the program MotionCor2 (ref. [44]) with 2× binning, yielding a pixel size of 1.06 Å. A sum of all frames of each image stack (50 in total) was calculated following a dose-weighting scheme and used for all image-processing steps except for defocus determination. The program CtfFind4 (ref. [45]) was used to estimate defocus values of the summed images from all movie frames. Particles were autopicked in Relion 3.1 (ref. [46]). After manual inspection and sorting to discard poor images, classifications were done in Relion 3.1. A total of 3,224,513 particles was extracted and subjected to one round of reference-free 2D classification to remove false picks and obvious junk classes. To speed up 3D classification during data processing, the entire dataset was divided into four batches in the order of their collection, and each batch was subjected to 3D classification individually. Only one class of each batch showed protein features and particles from this class were combined for further classification (1,007,363 particles in total). Auto-refinement was done on this particle set using the reconstruction from previous 3D classification as initial model and a soft mask surrounding the protein and detergent micelle. After this round of refinement, particles were subjected to Bayesian polishing, followed by another round of auto-refinement and focused refinement using a mask encompassing the ligase complex and Fab. The refinement at this step yielded a 3.3 Å map. Using the angle assignments obtained after the focused refinement, a 1.8° local 3D classification (2 sigma and $T = 20$) with an adaptive mask was used to further classify the particles. A total of 795,444 particles from one class was selected and subjected to another round of auto-refinement. 3D classification (T30) without alignment, but with a mask, was used to further improve the quality of the map. After selection of 121,644 particles, a final round of auto-refinement followed by focused refinement using the adaptive mask yielded a map at 3.1 Å. Local resolutions were calculated with Resmap v1.1.5 (ref. [47]) and map sharpening was performed in Relion 3.1. All reported resolutions are based on gold-standard refinement procedures and the Fourier shell correlation = 0.143 criterion. Histograms of directional Fourier shell correlation curves and sphericity values were calculated with the 3DFSC server[48]. All software is supported by SBGrid[49].

## Structural model building, refinement and analysis

All models were built in Coot[50] and refined in PHENIX[51] using the 3.1 Å sharpened density map. For Pex2, Pex10 and Pex12, transmembrane helices of each protein could easily be identified and were initially built as poly-Ala. We then manually fitted transmembrane helices into the density and traced the backbone of transmembrane segments and the linkers between them in Coot using secondary structure predictions and bulky residues (such as Phe, Trp and Arg) as sign posts. The density map was of sufficient quality to assign rotamers for key residues. In cases in which the rotamer could not be assigned, the side chain was stubbed at the Cβ atom. Models were refined in real space without secondary structure restraints using PHENIX real_space_refine. Strong non-crystallographic symmetry constraints in PHENIX real_space_refine were used to immobilize the domain that was not being refined. Several iterations of manual refinement and global refinement using Phenix and Coot were performed after visual inspection. For the ring finger domains, homology models of each RING domain were generated using

RaptorX[52] or Alphafold 2 (ref. [53]). Then, these models were fit into the density map in UCSF Chimera[54] and transferred to Coot for manual model building using secondary structure predictions from the XtalPred server as an additional guide. For Fab model building, the Fab portion of the deposited GgMFSD2A Fab complex (PDB ID: 7MJS) was used as a starting template and manually docked into the cryo-EM density with Chimera. The model was refined by iterative rounds of automated refinement. The structure of the Fab constant domain was removed due to its weak density. For models of lipids, PDB files of ergosterol or POPC were imported into Coot and fit into the density as ligands.

The Pex12 RF crystal dataset was collected at a wavelength of 0.967 Å, at which $Zn^{2+}$ has a strong anomalous signal. This signal gave good-quality anomalous data that allowed SHELXD to locate zinc atoms in a straightforward manner. Phase probability distributions using this dataset and heavy atom sites were calculated with the SHARP program[55]. The quality of the derived phases allowed most of the Pex12 RF model to be automatically or manually completed in Coot. Refinement was carried out with REFMAC[56].

Visualizations of the atomic models were made using UCSF Chimera[54], ChimeraX[57] and PyMOL (The PyMOL Molecular Graphics System, version 2.0, Schrödinger, LLC.).

### Peroxisome protein import assay

The violacein pathway (VioA, VioB and VioE-SKL)[19] was integrated into *S. cerevisiae* cells at the Leu2 locus. In brief, the plasmids pWCD1401 or pWCD1402 (ref. [58]) were digested with NotI-HF and the gel-extracted 11.5-kB fragments were used for transformation. Clones were selected on SD-Leu plates. The strain containing the violacein pathway was used to generate knockouts of Pex2, Pex10 and Pex12, as described above.

To complement strains lacking Pex2, Pex10 or Pex12, FLAG-tagged versions of the wild-type proteins or of mutants were expressed from the endogenous locus of each gene. The genes were introduced by homologous recombination using the vector pFA6A-HygMX, as described above. The plasmids were transformed into cells containing the violacein pathway, but lacking a PEX gene, and selected on SD-Leu medium. Three colonies from each transformation were streaked on a SD-Leu plate, and a single colony from each was picked for overnight growth in YPD medium at 30 °C with shaking at 250 r.p.m. Saturated cultures were then diluted 50-fold into 3 ml of fresh SD-Leu medium and grown for about 60 h.

Extraction of the green pigment PDV was done as follows. The cell pellet was resuspended in 300 μl of glacial acetic acid and transferred to thin-walled Eppendorf tubes. The tubes were then incubated at 95 °C for 15 min, mixed by inversion and incubated for another 15 min. Cell debris were removed first by centrifugation for 5 min at 4,700 r.p.m. and then by filtration of the supernatant with an Acroprep Advance 0.2-μm filter plate (Pall Corporation). The filtrate was transferred to a 96-well non-transparent plate (Greiner Bio-one). Fluorescence was determined with a microplate reader (Bio-Tek Synergy Neo2), using excitation and emission wavelengths of 535 nm and 585 nm, respectively[19].

### In vitro polyubiquitylation assay

In vitro polyubiquitylation assays were performed in reaction buffer (25 mM HEPES pH 7.4, 150 mM NaCl, 10 mM $MgCl_2$ and 50 μM TCEP) at 30 °C. The concentrations of the protein components were: 0.1 μM Uba1, 4 μM Ubc4, 0.5 μM GST–RFs and 100 μM ubiquitin. The mixture also contained 1 μM Dylight-Maleimide-800-labelled Cys-ubiquitin. The reaction was started by addition of 5 mM ATP and terminated after 60 min by addition of 4× SDS sample buffer. The samples were analysed by 4–20% SDS–PAGE and fluorescence scanning at 800 nm with an Odyssey scanner (Li-Cor).

### Quantitative isobaric tag-based proteomics

The samples were prepared and analysed by liquid chromatography–tandem mass spectrometry, as previously described[59]. In brief, yeast cells were cultured in 50 ml YNBG medium (0.3% yeast extract, 0.5% peptone, 0.67% yeast nitrogen base with amino acid and 0.5% glucose, pH 6.0) overnight until log phase and then switched into YNBO medium (0.3% yeast extract, 0.5% peptone, 0.67% yeast nitrogen base without amino acid, 0.5% glucose, 0.05% Tween40 and 0.1% oleic acid, pH 6.0) for another 18 h to induce peroxisome proliferation. Cells pellets were resuspended in lysis buffer (8 M urea, 200 mM EPPS pH 8.5 and protease inhibitors (Pierce)) and then lysed using a BioSpec Beadbeater for five cycles, 30 s on followed by 60 s off per cycle. The homogenate was centrifuged at 2,000g for 10 min to remove the cell debris and the supernatant was transferred to a new tube. The sample was reduced with 5 mM TCEP for 30 min, alkylated with 10 mM iodoacetamide for 30 min and then quenched with 10 mM DTT for 15 min. Streamlined tandem mass tag labelling and liquid chromatography–mass spectrometry were all done following the protocol described in ref. [59]. Quantitative isobaric tag-based proteomics data processing was done with MSconvert 3.0 (https://proteowizard.sourceforge.io/tools/msconvert.html) and Comet 2021.02 rev. 0 (http://comet-ms.sourceforge.net/).

### Immunoblotting

Yeast cells lacking Pex2 and Pex12 and expressing FLAG-tagged Pex2 and Pex12 wild-type or mutant proteins from the native locus under the endogenous promoter were cultured in 50 ml YNBG medium overnight until log phase. The cells were transferred into YNBO medium for another 18 h to induce peroxisome proliferation. Cells were lysed using glass beads and cell debris were removed by centrifugation, as described above. Membrane fractions were isolated by ultracentrifugation at 45,000 r.p.m. for 60 min. Membranes were homogenized and solubilized in lysis buffer containing 1% Triton X-100 for 1 h. The extract was then incubated with 10 μl of anti-FLAG M2 resin for 2 h at 4 °C. The beads were washed three times with lysis buffer containing 0.1% Triton X-100, and bound proteins were eluted with buffer containing 0.4 mg ml$^{-1}$ of 3×FLAG peptide (Sigma). Eluted proteins were subjected to SDS–PAGE and immunoblotting. FLAG-tagged Pex2 and Pex12 were detected using anti-FLAG (F7425, Sigma) antibodies at 1:3,000 dilution. As a loading control, total cell lysates were immunoblotted with anti-Sec61α antibody (homemade rabbit serum) at 1:3,000 dilution.

### Statistics and reproducibility

All biochemical experiments were independently performed at least three times with similar results. A one-way analysis of variance with multiple comparisons was performed using GraphPad Prism 9.3.0 to evaluate the statistical significance of peroxisomal protein import efficiency in wild-type cells compared to import in mutants of Pex2, Pex10, Pex12 or Pex5. The bar graphs shown in Figs. 3d and 4c–g and Extended Data Figs. 5h and 8c,d show the individual data points, the mean and s.e.m. from three biological repeats. NS, not significant; $*P < 0.1$, $**P < 0.05$, $****P < 0.001$. The quantitative isobaric tag-based proteomics experiment was performed independently twice with similar results. The statistics of results (Extended Data Fig. 9) were performed by multiple unpaired $t$-test followed by the method of two-stage step-up (Benjamini, Krieger and Yekutieli, desired false discovery rate ($Q$) = 1%) based on three biological replicates.

### Reporting summary

Further information on research design is available in the Nature Research Reporting Summary linked to this paper.

## Data availability

The cryo-EM density map and corresponding coordinates of the *T. thermophilus* Pex2, Pex10, Pex12 and Fab complex have been deposited in the Electron Microscopy Data Bank (EMDB) and the PDB under accession codes EMD-25750 and 7T92, respectively. The coordinates and crystallographic structure factors for *S. cerevisiae* RF12 were

deposited in the PDB under the accession code 7T9X. The mass spectrometry proteomics data have been deposited in the ProteomeXchange Consortium via the PRIDE partner repository with the dataset identifier PXD031792 (https://www.ebi.ac.uk/pride/). The structures of the two homodimeric RING domains (RNF4 and BIRC7) bound to their corresponding E2–Ub conjugates used for alignment are available in the PDB under the accession codes 4AP4 and 4AUQ, respectively. The structure of the GgMFSD2A–Fab complex used for Fab model building is available in the PDB under the accession code 7MJS. Uncropped versions of all gels and immunoblots are shown in Supplementary Fig. 1. Source data are provided with this paper.

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

**Acknowledgements** We thank Z. Yu, S. Yang and R. Yan at the HHMI Janelia Cryo-EM facility for help in microscope operation and data collection, the SBGrid team for software and workstation support, R. Erdmann and B. Gardner for providing materials, Y. Gao for help with the import assay, and Y. Gao, M. Yip, S. Shao and particularly M. Skowyra for comments on the manuscript. This work was supported by a NIGMS grant (R01 GM052586) to T.A.R., a NIH/NIGMS grant (R01 GM132129) to J.A.P. and by an NIGMS grant (R01 GM067945) to S.P.G. T.A.R. is a Howard Hughes Medical Institute Investigator. This article is subject to HHMI's Open Access to Publications policy. HHMI laboratory heads have previously granted a non-exclusive CC BY 4.0 license to the public and a sublicensable license to HHMI in their research articles. Pursuant to those licenses, the author-accepted manuscript of this article can be made freely available under a CC BY 4.0 license immediately upon publication.

**Author contributions** P.F. performed protein purifications, crystallization, ubiquitylation and import assays, and mutagenesis. P.F. and X.W. performed cryo-EM data collection, data processing and model building. S.K.E., P.K. and A.A.K. generated and selected Fabs. J.A.P. and S.P.G. performed quantitative mass spectrometry. T.A.R. supervised the project. P.F., X.W. and T.A.R. wrote the draft of the manuscript.

**Competing interests** The authors declare no competing interests.

**Additional information**
**Correspondence and requests for materials** should be addressed to Peiqiang Feng or Tom A. Rapoport.

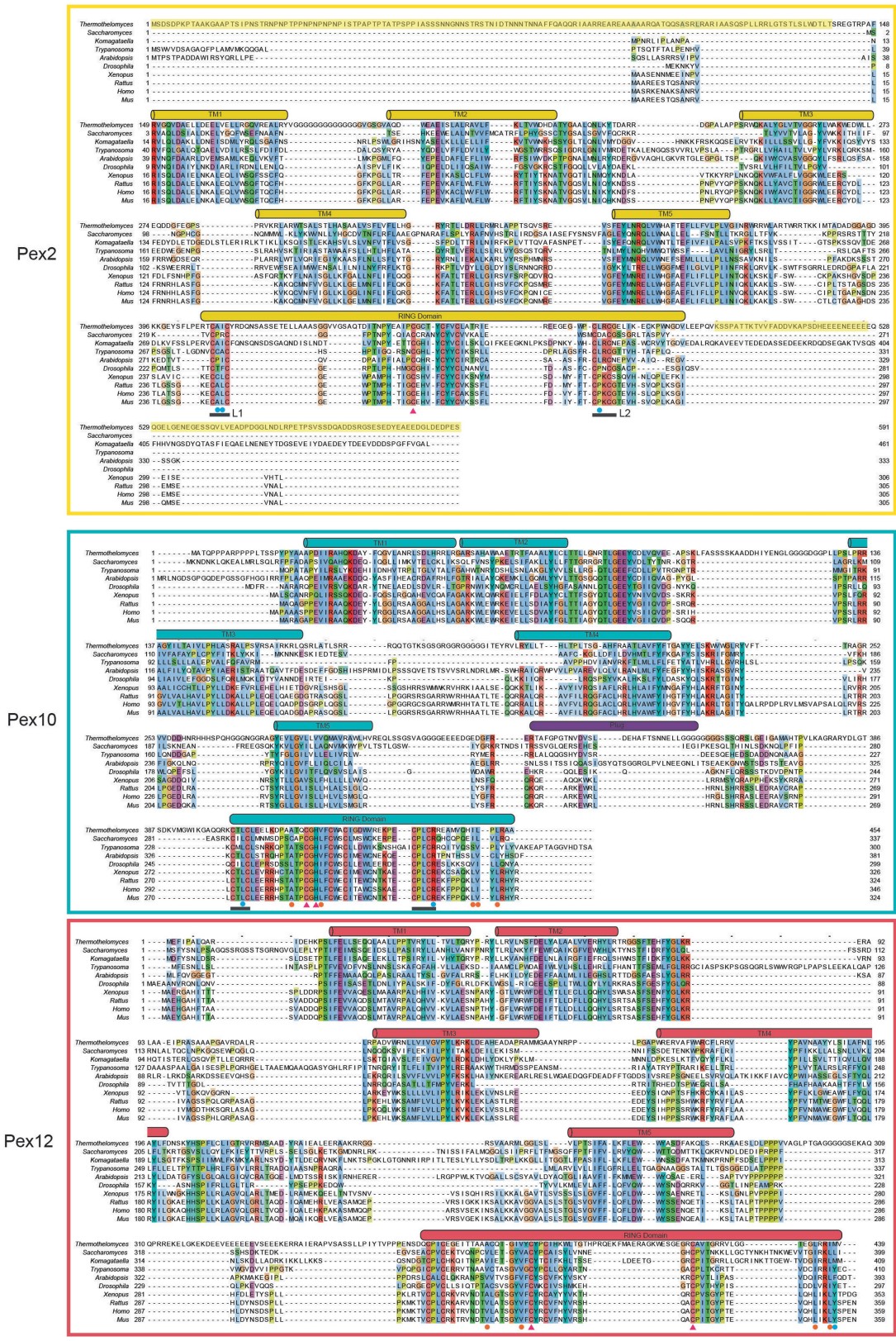

**Extended Data Fig. 1 | Sequence alignments of Pex2, Pex10, and Pex12 from different species.** All sequences were retrieved from Uniprot and aligned with the program MUSCLE[60], using the default parameters in JalView[61]. Amino acids were colored with ClustalX according to their properties. The degree of amino acid conservation is indicated by the intensity of the color. RFs are shown as round-cornered rectangles and TMs as cylinders, with boundaries according to the cryo-EM structure. The plug is shown in purple. Black lines indicate loops in RFs that interact with E2 enzymes; residues mutated in these loops are labeled by blue dots. Red triangles show Cys residues in RFs that were mutated. Orange dots highlight residues at the interface between RF10 and RF12. The N- and C-terminal Pex2 sequences shown with a yellow background were deleted in the *T. thermophilus* construct used for cryo-EM. The long isoform of Human PEX10 (UniProtKB – O60683-2) was used for sequence alignment.

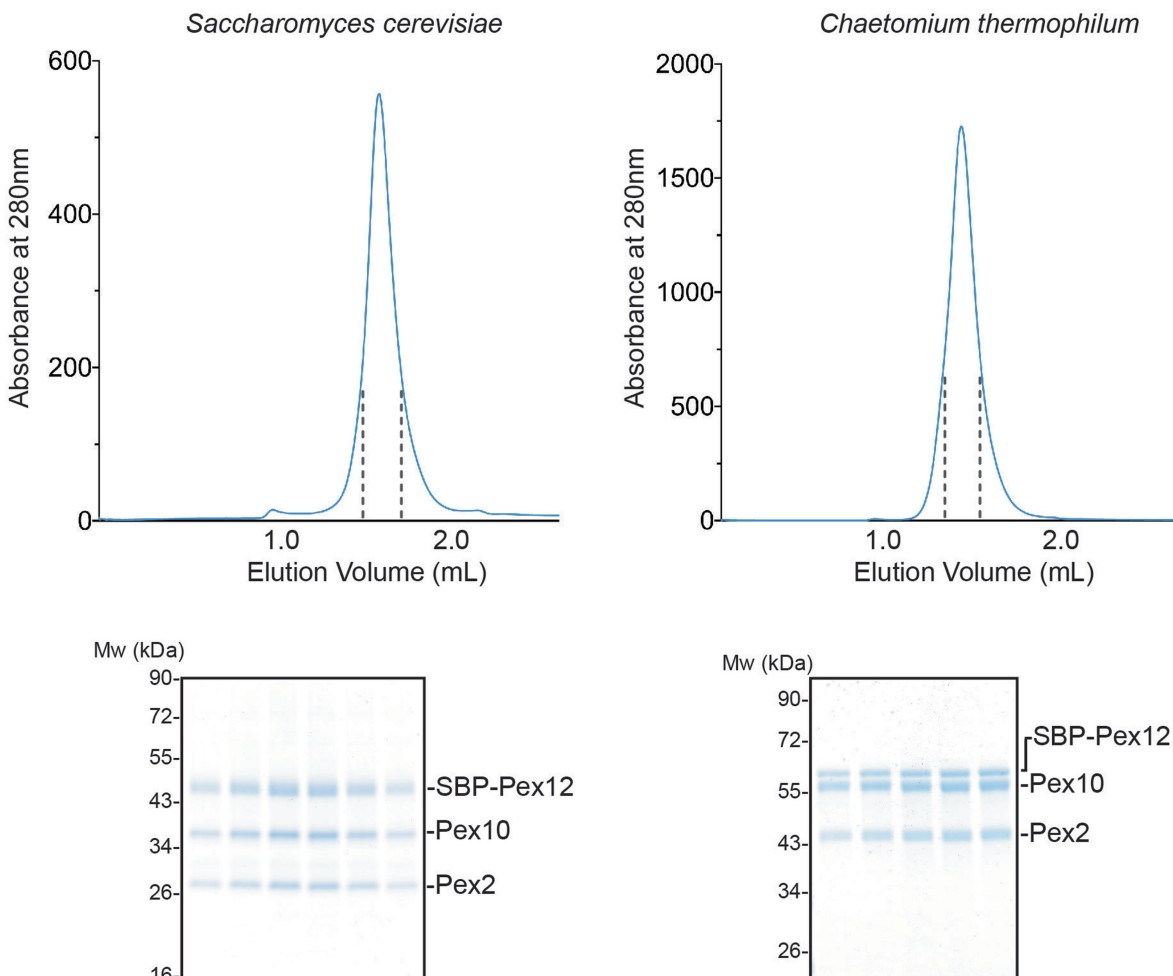

**Extended Data Fig. 2 | Pex2, Pex10, and Pex12 from *S. cerevisiae* or *C. thermophilum* form stoichiometric complexes.** Full-length *S. cerevisiae* Pex12 with an N-terminal streptavidin-binding peptide (SBP) tag was expressed together with full-length *S. cerevisiae* Pex10 and Pex2 in *P. pastoris*. The complex was purified with streptavidin beads and subjected to gel filtration. The upper panels show the absorbance profile. Fractions between the dashed lines were analyzed by SDS-PAGE and Coomassie blue staining (lower panels). The complex from *C. thermophilum* contained Pex2 (residues 107-455), lacking non-conserved residues at the N- and C-termini, and was purified in the same way. The results are representative of three biological repeats. For gel source data, see Supplementary Fig. 1.

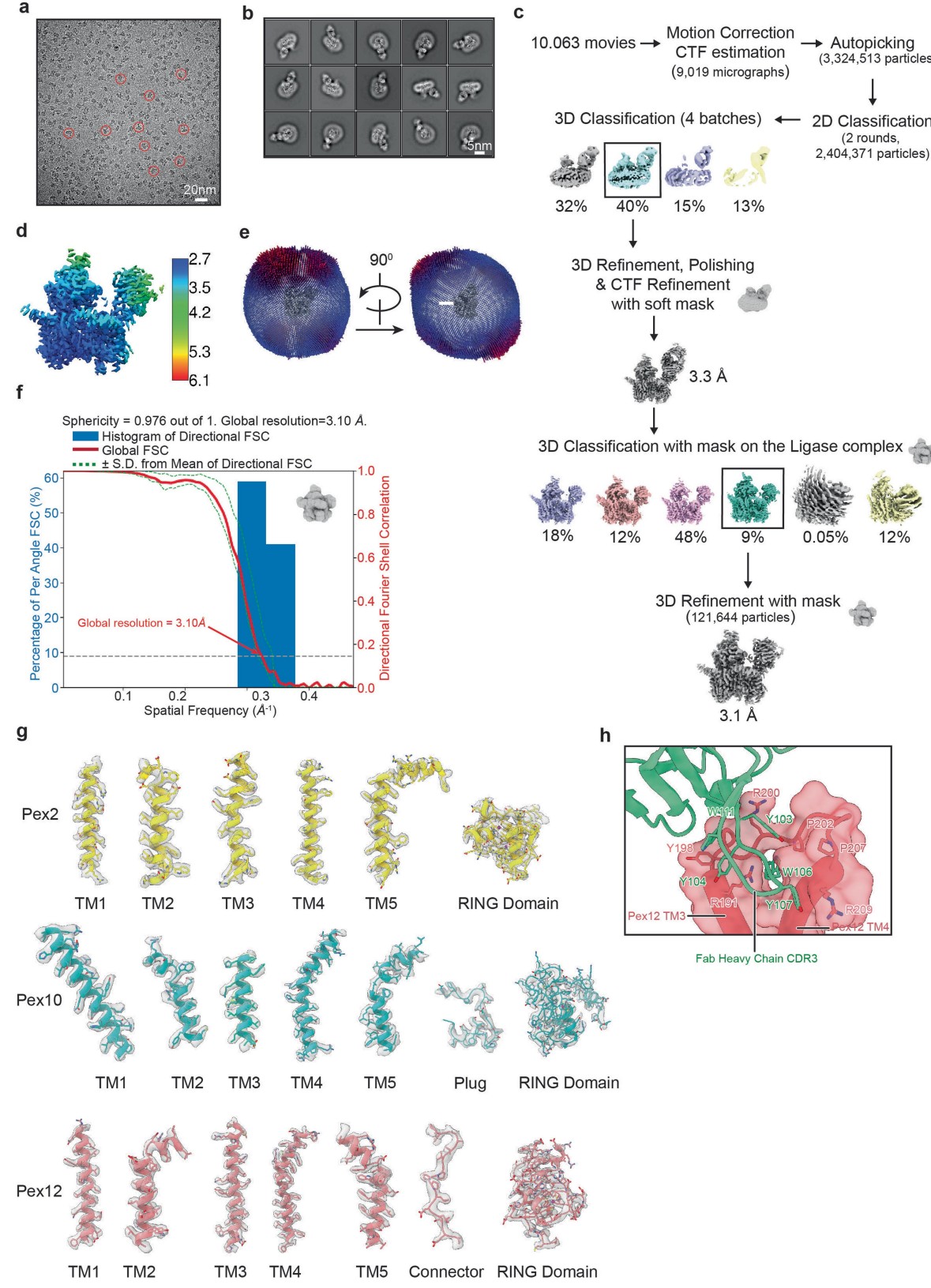

**Extended Data Fig. 3** | See next page for caption.

**Extended Data Fig. 3 | Cryo-EM data processing and reconstruction of the ubiquitin ligase complex with bound Fab. a**, Representative cryo-EM image. Similar results were obtained in two independent experiments. Selected particles are marked by red circles. **b**, Representative, reference-free 2D class averages of selected particles. **c**, Cryo-EM processing workflow (see also Extended Data Table 1). The classes selected for further analysis are boxed. Masks used for classification and refinement are indicated. **d**, Local resolution map with scale on the right. **e**, Euler angle distribution of refined particles shown in two different views. **f**, 3D Fourier shell correlation (FSC) curves and preferred orientation analysis. The red line shows the global FSC and the green dotted lines indicate the +1 and −1 standard deviations around the Mean of Directional FSC curve. The FSC calculations used the mask shown on the side. **g**, Density map and model for the TMs and RING domains (RFs) of Pex2, Pex10, and Pex12. The plug of Pex10 and the connector of Pex12 are also shown. **h**, Interaction of the Fab with a loop of Pex12. The Fv portion of the Fab is shown as a green cartoon. Pex12 is shown as a pink cartoon embedded in a space-filling model. Residues at the interface are highlighted.

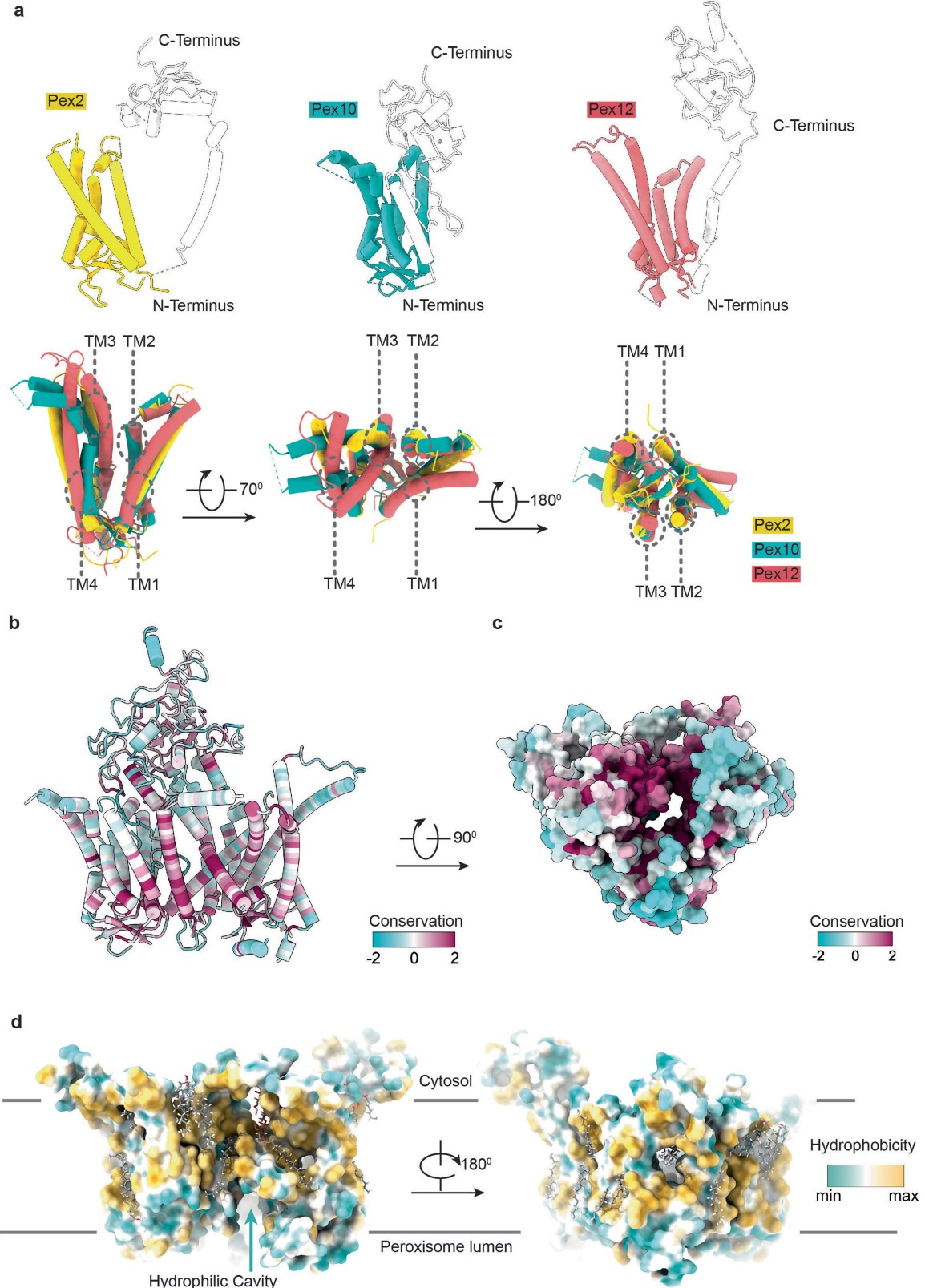

**Extended Data Fig. 4 | Structural relationship and amino acid conservation of the ligase complex components. a**, TMs1-4 of Pex2, Pex10, and Pex12 form similar structures. The upper panels show these TMs separately as cylinders, the lower panels show different views of their superposition. **b**, Shown is the degree of amino acid conservation among ligases of different species, as determined with the program ConSurf[62] (scale on the side). **c**, As in **b**, but conservation shown with a space-filling model viewed from the lumen. **d**, Space-filling model of the membrane-embedded domain viewed from the front and back with amino acids colored according to their hydrophobicity. A hydrophilic cavity is indicated by an arrow.

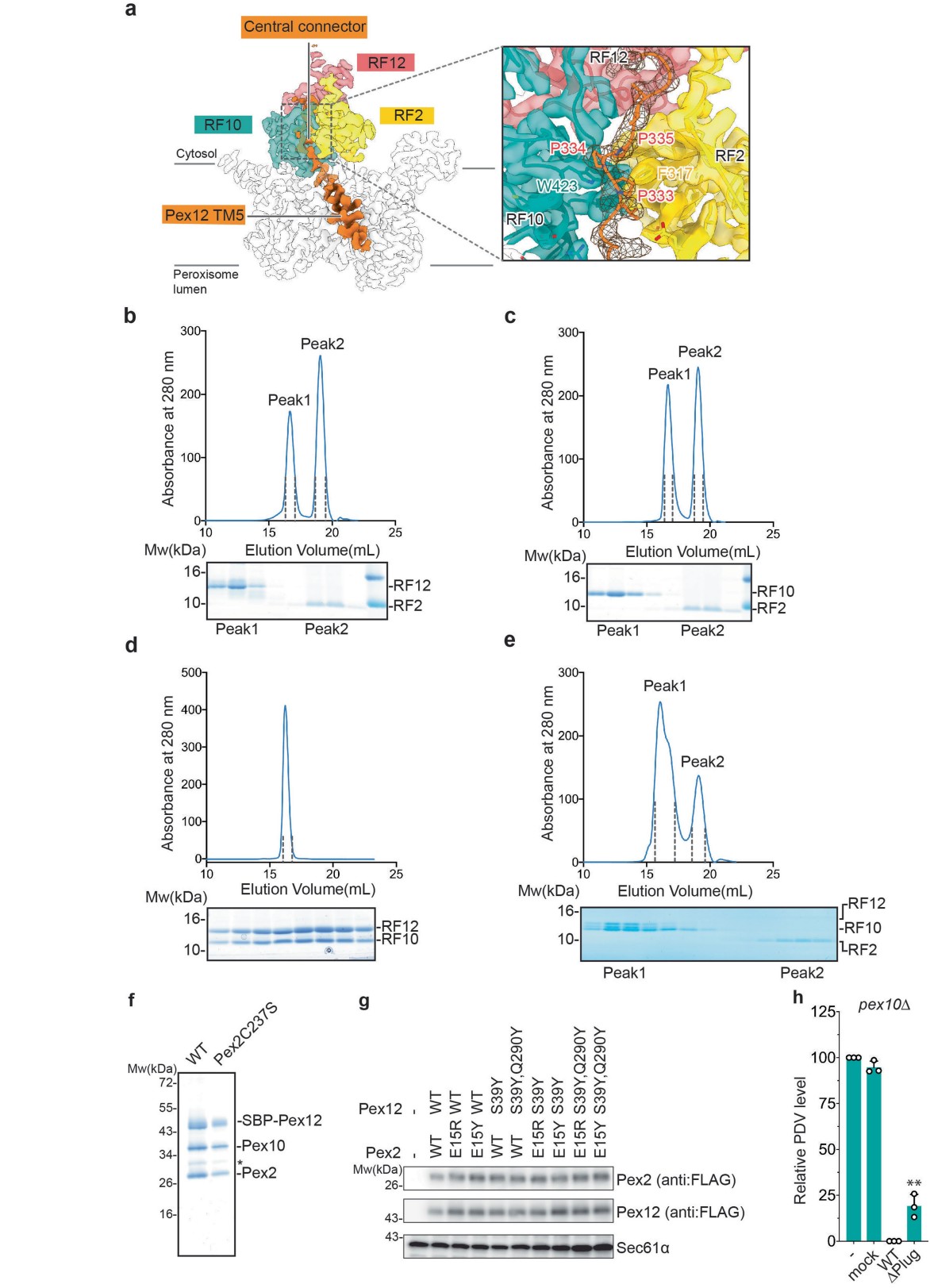

**Extended Data Fig. 5** | See next page for caption.

**Extended Data Fig. 5 | Testing interactions of the RFs and the roles of the plug and pore size. a**, Density map of the ligase complex, viewed from the side. The RF tower, TM5 of Pex12, and the central connector linking this TM to RF12 are shown in color. The right panel shows a magnified view of the boxed area. The conserved Pro residues in the central connector are indicated. **b**, Purified RF2 and RF12 were mixed and subjected to gel filtration. The upper panel shows the absorbance profile. Fractions between the broken lines were analyzed by SDS-PAGE and Coomassie blue staining (lower panel). **c**, As in **b**, but with a mixture of RF2 and RF10. **d**, As in **b**, but with a mixture of RF10 and RF12. **e**, As in **b**, but with a mixture of RF2, RF10, and RF12. **f**, Wild-type (WT) *S. cerevisiae* ligase complex and a complex containing mutant Pex2 (C237S), both containing Pex12 with an N-terminal streptavidin-binding peptide (SBP) tag, were expressed in *P. pastoris*. The complex was isolated with streptavidin beads and the bound material analyzed by SDS-PAGE and Coomassie blue staining. **g**, The expression levels of FLAG-tagged Pex2 or Pex12 mutants that reduce the pore size (see Fig. 3d) were determined by immunoblotting with FLAG antibodies. Blotting for Sec61α served as a loading control. The results in **b**–**g** are representative of three biological repeats. For gel source data, see Supplementary Fig. 1. **h**, Wild-type (WT) Pex10 or a mutant lacking the putative plug (Δplug) were expressed from the endogenous promoter in *S. cerevisiae* cells lacking Pex10 (*pex10Δ*). Controls were performed with *pex10Δ* cells and cells expressing only an antibiotic resistance gene (mock). Peroxisomal protein import was determined by the reduction in the formation of a fluorescent pigment (PDV). Fluorescence was measured in cell lysates and the data were normalized, setting the fluorescence of *pex10Δ* cells as 100% and that of WT cells as 0%. The bar graph shows the individual data points, the mean and S.E.M. from three biological repeats. Statistical significance between the wild type and mutants was calculated by one-way analysis of variance, $**P < 0.01$. See also Source Data file.

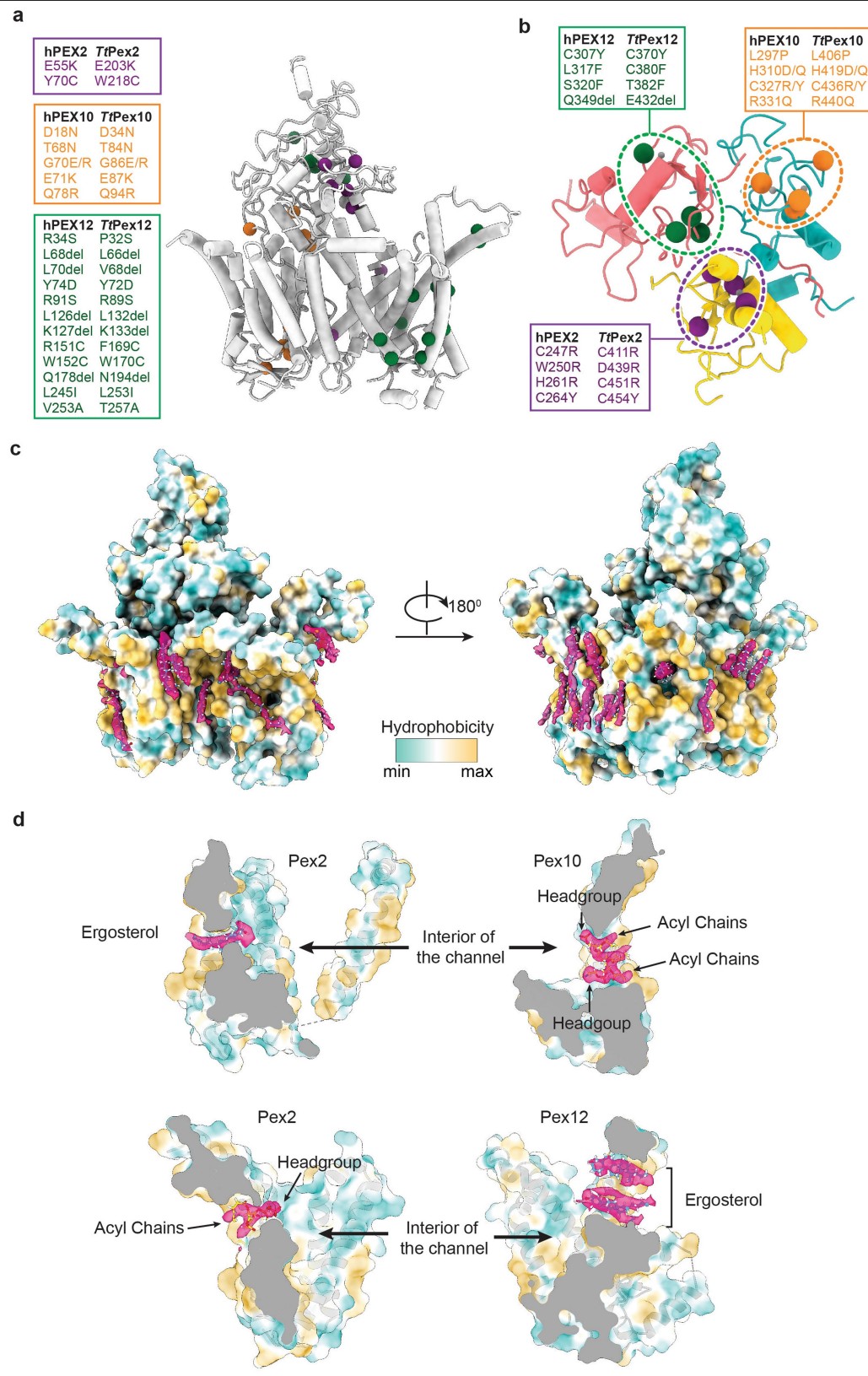

**Extended Data Fig. 6 | Location of disease-causing point mutations and of phospholipid and cholesterol molecules. a**, A side view of the model for the ligase complex, with helices as cylinders. Point mutations found in patients in PEX2, PEX10, or PEX12 are indicated as balls in different colors. The specific mutations and the corresponding positions in the *T. thermophilus* (*T.t.*) proteins are listed on the side. **b**, A magnified view of the RFs from the cytosol, with disease mutations highlighted. The long isoform of Pex10 (UniProtKB – O60683-2) was used for residue numbering. **c**, Space-filling model of the ligase complex in two different side views. Amino acids are colored according to the degree of their hydrophobicity (scale below the arrow). Cryo-EM density modeled as phospholipid or ergosterol molecules is colored in purple. **d**, Examples of phospholipid and ergosterol molecules plugging holes in the channel walls. Shown is a cut through the semi-transparent space-filling and embedded cartoon models. For phospholipids, the modeled head groups and acyl chains are indicated.

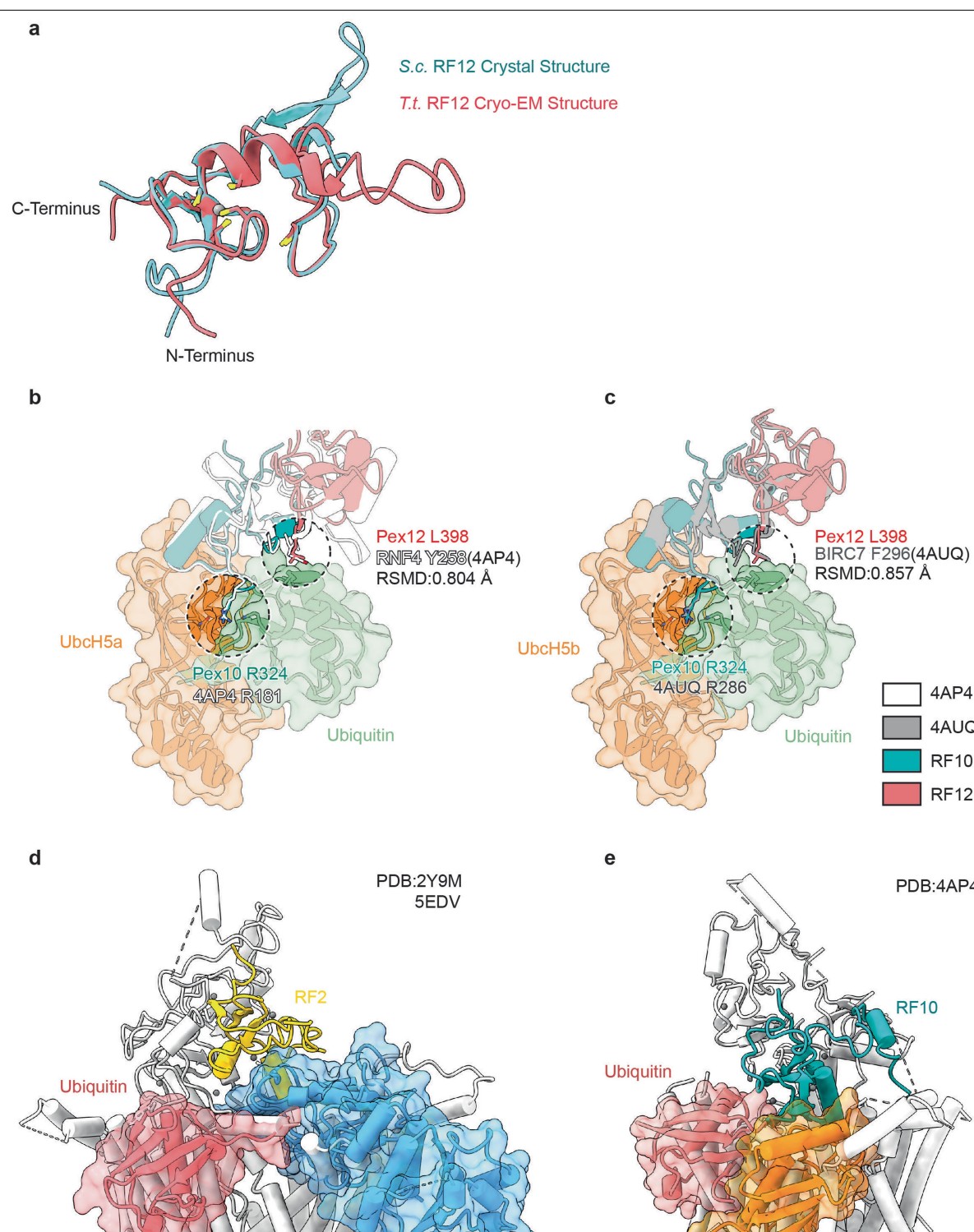

**Extended Data Fig. 7** | See next page for caption.

**Extended Data Fig. 7 | Structural features of RF12 and putative interactions of the RFs with E2 enzymes. a**, Overlay of the cryo-EM structure of *T. thermophilus* (*T.t.*) RF12 with the crystal structure of *S. cerevisiae* (*S.c.*) RF12 (see also Extended Data Table 2). The structures are shown as cartoons with Cys residues as yellow sticks and the bound $Zn^{2+}$ atom as a grey ball. Note that the cores of the structures are similar, but several loops are different. **b**, Putative interaction between the *S. cerevisiae* RF10–RF12 complex and an E2–Ub conjugate. The structure of RF10 is a homology model, based on the cryo-EM structure, and that of RF12 a crystal structure (**a**). The structure of RF10–RF12 was aligned with the structure of the homodimeric RF of the ubiquitin ligase RNF4 bound to the ubiquitin-conjugated E2 enzyme UbcH5a (PDB code 4AP4). The alignment is based on RF10 and one of the RFs in RNF4. UbcH5a and ubiquitin are shown as cartoons inside a semi-transparent space-filling model. RF10 and RF12 are shown as colored cartoons, and the RFs of RNF4 as white cartoons. One circle highlights the linchpin residues of RF10 and RNF4 (R324 and R181, respectively), the other shows the interaction between L398 of Pex12 or the equivalent residue in RNF4 (Y258) with the ubiquitin helix. **c**, As in **b**, but for the homodimeric RF ligase BIRC7 (PDB code 4AUQ). **d**, The modelled structure of an E2-Ub conjugate was docked onto the putative binding site of RF2. The observed clashes suggest that RF2 must undergo a conformational change for its activation. **e**, As in **d**, but for the docking of E2–Ub onto RF10. Severe clashes suggest again a conformational change for activation.

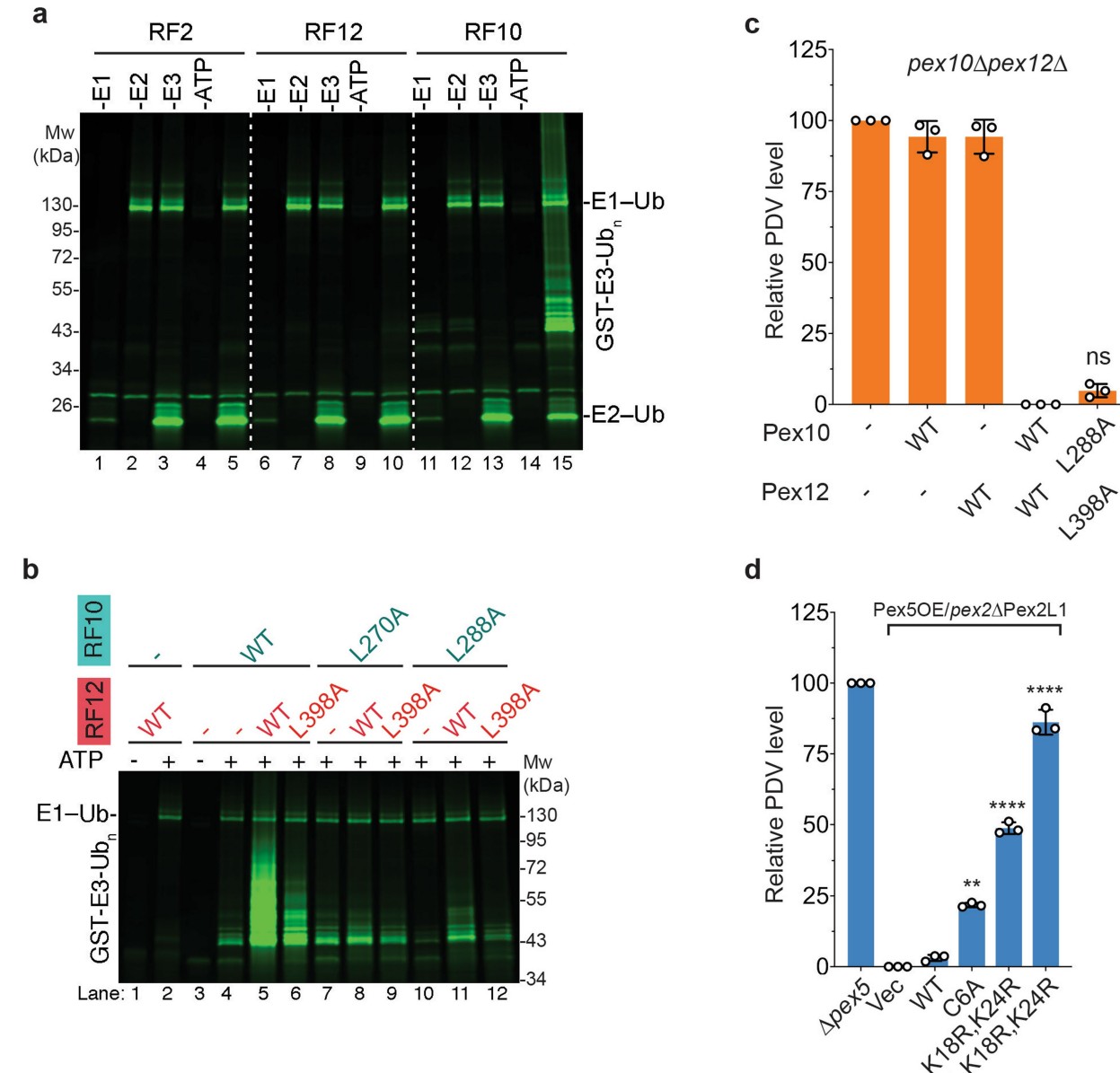

**Extended Data Fig. 8 | *In vitro* polyubiquitination activity of the RFs.**
**a**, Polyubiquitination reactions were performed with purified GST-fusions of
RF2, RF12, or RF10 (E3 enzyme) in the presence of E1 enzyme, the E2 enzyme
Ubc4, and Dylight800-labeled ubiquitin. Where indicated, components were
omitted. The samples were analyzed by SDS-PAGE and fluorescence scanning.
**b**, As in **a**, but with different combinations of wild-type (WT) or mutant RF10
and RF12. The results in **a** and **b** are representative of three biological repeats.
For gel source data, see Supplementary Fig. 1. **c**, WT Pex10 or Pex12, or the
indicated mutants, were expressed from endogenous promoters in
*S. cerevisiae* cells lacking Pex10 and Pex12 (*pex10Δ pex12Δ*). Peroxisomal
protein import was determined by the reduction in the formation of a

fluorescent compound (PDV). Fluorescence was measured in cell lysates and
the data were normalized, setting the fluorescence of *pex10Δ pex12Δ* cells as
100% and that of cells expressing WT Pex10 and Pex12 as 0%. Shown are the
means and standard errors of three experiments. **d**, WT or mutant Pex5 was
overexpressed (Pex5OE) in Pex2 mutant cells containing mutations in the L1
loop of RF2 (*pex2Δ* Pex2L1). Vec, vector without Pex5. Cells lacking Pex5
(*pex5Δ*) served as a control. Protein import was measured as in **c**. The bar
graphs in **c** and **d** show the individual data points, the mean and S.E.M. from
three biological repeats. Statistical significance between the wild type and
mutants was calculated by one-way analysis of variance. ns, not significant;
** $P < 0.05$; **** $P < 0.001$. See also Source Data file.

**a**

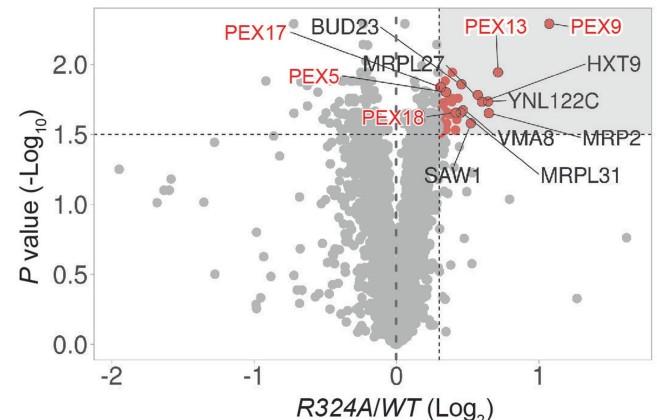

**b**

| Protein Name | Fold change (Log2) | P value(-Log10) |
|---|---|---|
| PEX9 | 1.073 | 2.291 |
| PEX13 | 0.713 | 1.943 |
| MRP2 | 0.649 | 1.652 |
| YNL122C | 0.644 | 1.735 |
| HXT9 | 0.600 | 1.733 |
| MRPL27 | 0.571 | 1.782 |
| SAW1 | 0.522 | 1.579 |
| VMA8 | 0.467 | 1.672 |
| BUD23 | 0.456 | 1.858 |
| MRPL31 | 0.451 | 1.655 |
| PEX18 | 0.417 | 1.655 |
| PEX5 | 0.349 | 1.803 |
| PEX17 | 0.313 | 1.837 |

**Extended Data Fig. 9 | Identification of substrates of the polyubiquitination pathway by quantitative isobaric tag-based proteomics. a**, Proteins from wild-type (WT) *S. cerevisiae* cells, or cells expressing a Pex10 mutant defective in polyubiquitination (R324A) under the endogenous promoter, were subjected to isobaric tag-based proteomics using mass spectrometric analysis. Results are representative of two independent experiments. Each dot in the volcano plot represents a protein for which the ratio of its abundance in the R324A mutant and in wild-type cells (R324A/WT) is given, as well as a measure of statistical significance (p-value). The statistics was performed by Multiple Unpaired *t*-test followed by the method of Two-Stage Step-up (Benjamini, Krieger, and Yekutieli, Desired FDR (Q) = 1%) based on three biological replicates. Proteins indicated by red dots in the upper rectangle significantly accumulate in the mutant strain ($P < 0.03$ and R324A/WT >1.3). **b**, List of the proteins indicated as red dots in **a**. Peroxisomal proteins are highlighted with a red background. The other listed proteins are probably indirectly affected by the Pex10 mutation or erroneously identified (they include mitochondrial ribosomal proteins, cell wall proteins, and DNA binding proteins).

**Extended Data Table 1 | Cryo-EM data collection, refinement, and validation statistics for the *T.t.* ligase complex with bound Fab**

|  | *T.t.* Ligase complex with Fab bound (EMD-25750) (PDB 7T92) |
| --- | --- |
| Data collection and processing |  |
| Magnification | 81,000 |
| Voltage (kV) | 300 |
| Electron exposure (e–/Å$^2$) | 52.83 |
| Defocus range (µm) | -1.0 to -2.5 |
| Pixel size (Å) | Super-resolution 0.53 |
| Symmetry imposed | C1 |
| Initial particle images (no.) | 3,324,513 |
| Final  particle images (no.) | 121,644 |
| Map resolution (Å) |  |
| FSC threshold | 0.143 |
| Map resolution range (Å) | 2.7 to 5.0 |
|  |  |
| Refinement |  |
| Initial model used (PDB code) |  |
| Model resolution (Å) | 3.1 |
| FSC threshold | 0.5 |
| Model resolution range (Å) | 2.7 to 5.0 |
| Map sharpening B factor (Å$^2$) | -80 |
| Model composition |  |
| Non-hydrogen atoms | 9673 |
| Protein residues | 1159 |
| Ligands | 30 |
| B factors (Å$^2$) |  |
| Protein | 39.91 |
| Ligand | 31.67 |
| R.m.s. deviations |  |
| Bond lengths (Å) | 0.005 |
| Bond angles (°) | 0.729 |
| Validation |  |
| MolProbity score | 1.61 |
| Clashscore | 7.11 |
| Poor rotamers (%) | 0 |
| Ramachandran plot |  |
| Favored (%) | 96.59 |
| Allowed (%) | 3.41 |
| Disallowed (%) | 0 |

**Extended Data Table 2 | Data collection and refinement statistics for the crystal structure of the S.c. Pex12 RING domain**

|  | *S.c.* Pex12 RING domain (PDB 7T9X) |
|---|---|
| **Data collection** | |
| Space group | P 1 21 1 |
| Cell dimensions | |
| $a, b, c$ (Å) | 27.56, 32.91, 72.43 |
| $\alpha, \beta, \gamma$ (°) | 90.00, 100.72, 90.00 |
| Resolution (Å) | 1.56* |
| $R_{sym}$ or $R_{merge}$ | 0.057 |
| $I / \sigma I$ | 36.1 |
| Completeness (%) | 97.1 |
| Redundancy | 9.3 |
|  | |
| **Refinement** | |
| Resolution (Å) | 1.52 |
| No. reflections | 18532 |
| $R_{work} / R_{free}$ | 0.186/0.197 |
| No. atoms | 1190 |
| Protein | 1114 |
| Ligand/ion | 2 |
| Water | 74 |
| $B$-factors | |
| Protein | 17.32 |
| Ligand/ion | 13.59 |
| Water | |
| R.m.s. deviations | |
| Bond lengths (Å) | 0.005 |
| Bond angles (°) | 0.838 |

The dataset was collected from one single crystal. *Values in parentheses are for highest-resolution shell.

# Reporting Summary

## Statistics

For all statistical analyses, confirm that the following items are present in the figure legend, table legend, main text, or Methods section.

| n/a | Confirmed | |
|---|---|---|
| ☐ | ☒ | The exact sample size (*n*) for each experimental group/condition, given as a discrete number and unit of measurement |
| ☐ | ☒ | A statement on whether measurements were taken from distinct samples or whether the same sample was measured repeatedly |
| ☐ | ☒ | The statistical test(s) used AND whether they are one- or two-sided *Only common tests should be described solely by name; describe more complex techniques in the Methods section.* |
| ☒ | ☐ | A description of all covariates tested |
| ☒ | ☐ | A description of any assumptions or corrections, such as tests of normality and adjustment for multiple comparisons |
| ☐ | ☒ | A full description of the statistical parameters including central tendency (e.g. means) or other basic estimates (e.g. regression coefficient) AND variation (e.g. standard deviation) or associated estimates of uncertainty (e.g. confidence intervals) |
| ☐ | ☒ | For null hypothesis testing, the test statistic (e.g. *F*, *t*, *r*) with confidence intervals, effect sizes, degrees of freedom and *P* value noted *Give P values as exact values whenever suitable.* |
| ☒ | ☐ | For Bayesian analysis, information on the choice of priors and Markov chain Monte Carlo settings |
| ☒ | ☐ | For hierarchical and complex designs, identification of the appropriate level for tests and full reporting of outcomes |
| ☒ | ☐ | Estimates of effect sizes (e.g. Cohen's *d*, Pearson's *r*), indicating how they were calculated |

*Our web collection on statistics for biologists contains articles on many of the points above.*

## Software and code

Policy information about availability of computer code

| | |
|---|---|
| Data collection | Cryo-EM data collection was done on a Titan Krios (Thermo Fisher Scientific) electron microscope (300keV) equipped with a Gatan K2 direct electron detector (Gatan). Automatic data collection was done with SerialEM v.3.8-beta. Negative staining EM data collection was done on a Thermo Scientific Tecnai T12 equipped with a Gatan UltraScan 895 (4k x 4k) CCD camera. Crystal data collection was done at APS beamlines 23-ID-C. HKL2000/3000 packages were used for data processing. Quantitative isobaric tag-based proteomics was done on the Orbitrap Lumos Mass Spectrometer (ThermoFisher Scientific) with Proxeon NanoLC-1200 UHPLC (ThermoFisher Scientific) and Accucore Columns (ThermoFisher Scientific). |
| Data analysis | Cryo-EM and Negative staining EM data processing were done with RELION v.3.1.0, UCSF MotionCor2 (MotionCor2_1.1.0-Cuda80), CTFFIND v.4.1, UCSF Chimera v.1.14, and UCSF ChimeraX v.1.2, PyMOL 2.4.1. Model building was done using Coot 0.9.4, Phenix 1.19.2 and CCP4 7.1.011. Local resolutions were calculated with Resmap v1.1.5. Histograms of directional FSC curves and sphericity values were calculated with the 3DFSC Program Suite Version 3.0. Quantitative isobaric tag-based proteomics data processing was done with MSconvert 3.0 (https://proteowizard.sourceforge.io/tools/msconvert.html) and Comet 2021.02 rev. 0 (http://comet-ms.sourceforge.net/). |

For manuscripts utilizing custom algorithms or software that are central to the research but not yet described in published literature, software must be made available to editors and reviewers. We strongly encourage code deposition in a community repository (e.g. GitHub). See the Nature Portfolio guidelines for submitting code & software for further information.

## Data

Policy information about availability of data

All manuscripts must include a data availability statement. This statement should provide the following information, where applicable:
- Accession codes, unique identifiers, or web links for publicly available datasets
- A description of any restrictions on data availability
- For clinical datasets or third party data, please ensure that the statement adheres to our policy

The cryo-EM density map and corresponding coordinate of the Thermothelomyces thermophilus Pex2, Pex10, Pex12, Fab complex have been deposited in the Electron Microscopy Data Bank (EMDB) and Protein Data Bank (PDB) under accession codes EMD-25750 and PDB 7T92, respectively. The coordinates and crystallographic structure factors for Saccharomyces cerevisiae RF12 were deposited in the Protein Data Bank (PDB) under the accession code PDB 7T9X. The mass spectrometry proteomics data have been deposited to the ProteomeXchange Consortium via the PRIDE partner repository with the dataset identifier PXD031792 (https://www.ebi.ac.uk/pride/). The structures of the two homodimeric RING domains (RNF4 and BIRC7) bound with corresponding E2~Ub conjugates are under the accession codes PDB 4AP4 and PDB 4AUQ in the Protein Data Bank (PDB), respectively. The structure of GgMFSD2A Fab complex for Fab model building during structure determination is under the accession code PDB 7MJS in the Protein Data Bank (PDB). Uncropped version of all the gels and immunoblot results are included as Supplementary Fig. 1. Source data are provided with this paper.

# Field-specific reporting

Please select the one below that is the best fit for your research. If you are not sure, read the appropriate sections before making your selection.

☒ Life sciences  ☐ Behavioural & social sciences  ☐ Ecological, evolutionary & environmental sciences

For a reference copy of the document with all sections, see nature.com/documents/nr-reporting-summary-flat.pdf

# Life sciences study design

All studies must disclose on these points even when the disclosure is negative.

| | |
|---|---|
| Sample size | No statistical methods were used to predetermine sample size. All functional data were obtained from at least three independent experiments to ensure each data points was repeatable and comparable to other published studies. The amount of proteins for in vitro biochemical experiments was chosen based on previous experience with this specific type of experiments and commonly used sample sizes in the field of research. For single particle cryo-EM reconstruction, sample sizes were determined by available electron microscopy time and the number of particles on each micrograph obtained during the collection time. |
| Data exclusions | No data were excluded from our analysis. |
| Replication | The number of replications for each experiment is stated in the Figure Legends. |
| Randomization | No randomization was performed, since this study did not allocate experimental groups. |
| Blinding | Blinding is not relevant to this study, as no subjective allocation was involved in any of the structural and functional experiments. |

# Reporting for specific materials, systems and methods

We require information from authors about some types of materials, experimental systems and methods used in many studies. Here, indicate whether each material, system or method listed is relevant to your study. If you are not sure if a list item applies to your research, read the appropriate section before selecting a response.

## Materials & experimental systems

| n/a | Involved in the study |
|---|---|
| ☐ | ☒ Antibodies |
| ☐ | ☒ Eukaryotic cell lines |
| ☒ | ☐ Palaeontology and archaeology |
| ☒ | ☐ Animals and other organisms |
| ☒ | ☐ Human research participants |
| ☒ | ☐ Clinical data |
| ☒ | ☐ Dual use research of concern |

## Methods

| n/a | Involved in the study |
|---|---|
| ☒ | ☐ ChIP-seq |
| ☒ | ☐ Flow cytometry |
| ☒ | ☐ MRI-based neuroimaging |

## Antibodies

| | |
|---|---|
| Antibodies used | Anti-FLAG antibody produced in rabbit (Sigma #F7425, dilution=1:3000), anti-Sec61 antibody (lab homemade stock, dilution=1:3000). |

| Validation | The Anti-FLAG antibody from Sigma was validated using immunoblotting as described on the product's website(https://www.sigmaaldrich.com/US/en/product/sigma/f7425). The anti-Sec61 antibody was validated using immunoblotting as described in previous study (Panzner, S., et, al. 1995. Cell). |
|---|---|

## Eukaryotic cell lines

Policy information about cell lines

| Cell line source(s) | Pichia pastoris SMD1168 (Invitrogen-C17500), Saccharomyces cerevisiae UTL7A (gift from Dr. Ralf Erdmann) |
|---|---|
| Authentication | All cell lines were authenticated with positive control following manufacture's instruction or provided protocol. |
| Mycoplasma contamination | Mycoplasma contamination is not applicable to this study. |
| Commonly misidentified lines (See ICLAC register) | We did not use any commonly misidentified cell line. |

