## [Peer Review File · Nature]

Manuscript Title: A peroxisomal ubiquitin ligase complex forms a retro-translocation channel.

Editorial Notes:

Redactions – unpublished data

Reviewer Comments & Author Rebuttals

Reviewer Reports on the Initial Version:

Referees' comments:

Referee #1 (Remarks to the Author):

This is an elegant structure-function study of a novel membrane-associated E3 ligase Pex2/10/12 of the yeast peroxisome. The cryo-EM structure was determined to an average resolution of 3.1 Å, with the help of a protein surface PEGylation which likely improved the particle orientation distribution in the EM grids as well as the production of a Fab fragment that increased the size of the small complex (150 kDa) and facilitated particle image alignment. The cryo-EM analysis is well performed. The only weakness is the relatively low resolution of the largely flexible RF2 domain. But the authors have remediated the problem by determining a high-resolution crystal structure of the truncated domain. The structure is unexpected and striking, revealing a constitutive open channel of 10 Å formed by 15 transmembrane helices, five from each subunit. Mutagenesis and functional assays demonstrate that this aqueous channel is utilized by the N-termini of the recycling receptors from the peroxisome lumen to reach the catalytic RF domains on the cytosolic side of the peroxisome membrane for ubiquitination. Another key insight is the identification of the mono-ubiquitination and poly-ubiquitination mechanisms in Pex2/10/12 ligase complex. The manuscript is very well written, and the Figures are well prepared and illustrative. This work represents an important advance on peroxisome biogenesis that will be of interest to the wide audience of Nature.

Minor corrections

1. There is an unintended horizontal white line across the yellow histogram bars in Fig. 4e.
2. Page 7 line 149: The sentence does not sound logic and may need attention.

Referee #2 (Remarks to the Author):

This interesting paper by Feng et al. is a potentially exciting advance because the three RING-domain peroxins, Pex2, Pex10 and Pex12, residing in the peroxisomal membrane serve as E3 ligases for several aspects of peroxisome dynamics, such as peroxisome biogenesis, PTS receptor quality control (RADAR) and pexophagy. Structural information provided by this paper is therefore potentially interesting, particularly as it relates to the roles of these RING peroxins in PTS receptor mono- and poly-ubiquitination reactions. In this paper, the authors focus on the structure of the hetero-trimeric complex of Pex2, Pex10 and Pex12, primarily of fungal origin. Although several PTS receptors (Pex5, Pex7) and their co-receptors (Pex18/20/21) are ubiquitinated by this RING

subcomplex, the authors have studied only Pex5 ubiquitination physiologically and present models for how the RING subcomplex could be involved in mono- and poly-ubiquitination of targets during the peroxisomal matrix protein import cycle.

The strength of the paper is that it is structurally excellent with the determination of RING complex structures from several sources contributing to the analysis. It is also original. The data are of high quality, although the description of the methods is sub-optimal to engender reproducibility. Although the topic is of potential interest and the structural work is sound, several issues, including the interpretation of the data and their physiological relevance suggest that it is too preliminary at present and much further work would be needed.

1. The authors could not detect any mono-ubiquitination by the RING subcomplex, which normally uses the endogenous Pex4 as the E2 enzyme. Instead the authors used Ubc4 as the E2 enzyme for all their in vitro studies. Thus, a key aspect of the role of this RING subcomplex in PTS receptor recycling has not been studied or recapitulated. Although the authors present a model for how Pex2 might be involved in Pex5 mono-ubiquitination (lines 140147 and Fig. 4h), it is speculative, without data to back it up, and contradicts previous publications suggesting that Pex12 facilitates the Pex4-dependent mono-ubiquitination of Pex5 (Platta et al., 2009; PMID: 19687296).
2. The authors have studied Pex5 poly-ubiquitination (which is involved in the RADAR pathway) and present a nice model suggesting that Pex10, stimulated by Pex12, does this poly-ubiquitination. However, this too contradicts a previous report suggesting that Pex2 mediates the Ubc4-dependent polyubiquitination (Platta et al., 2009; PMID: 19687296), although it does agree with another report showing that Pex10 is the E3 ligase involved in Ubc4-dependent poly-ubiquitination of Pex5 (Williams C et al 2008; PMID: 18644345).
3. It is unclear from the manuscript what the site/s of poly-ubiquitination are in Pex5, which is important because the mono-ubiquitination of Pex5 is at a Cys near the N-terminus, but the poly-ubiquitination is at Lys residues elsewhere (Lys18 and Lys24 in yeast Pex5) and the physiological consequences of modifications at these sites are very different.
4. The authors do not present their data carefully – one example is that many sites of mutation in human PEX10 (and other proteins) are presented incorrectly in the text and one Figure. As examples, the reference to H310D/Q is incorrectly cited in hPEX10, as is L297 and C327 (Fig. 7b). Extended Data Fig. 7b has several incorrectly labeled residues in hPEX10 and likely TtPEX10. The authors should check all the numbering of residues from all species. In Fig. 4d, should C353S be C354S?
5. An unexpected surprise is that this paper claims that all the RING peroxins have 5 transmembrane domains (TMDs). Given that there are contradictory reports in the literature suggesting otherwise - e.g. Pex12 is reported to have either an even (N and C- termini of PEX12 face the cytosol, Okumoto et al., 1998; PMID: 9632816), or odd (Albertini et al., 2001; PMID: 11370741) number of TMDs - it is obligatory that the authors provide biochemical evidence for the membrane topology and orientation of the termini, minimally for at least one of the RING peroxins.
6. The authors should cite that PEX2 is a E3 ligase for mammalian pexophagy (Sargent et al., 2016; PMID: 27597759; because this points out that there are other targets, such as PMP70, and a different site, Lys209, on PEX5 for mono-ubiquitination).
7. Open pore discussion starting line 127 – In peroxisomes, it is proposed that Pex8 is part of the importomer (Agne et al., 2003; PMID: 12667447) and that it may be involved in cargo release (Ma et al., 2013; PMID: 23902771). The authors should use caution to call it an “open pore” before ruling out that it might be occluded in vivo prior to cargo release.
8. In the model in Fig. 4h, where does cargo release fit in? Without this step, what would prevent Pex5 bound PTS cargo from being exported? Also, the statement about the pore being constitutively “open” is speculation at this stage (lines 262,263 – see previous comment regarding Pex8).

Minor

The authors use multiple names and spellings for *T. thermophilus* (including *myceliophthra* in Extended Data Fig. 1)

There is inconsistent use of yeast and mammalian nomenclature for peroxins (Pex# for yeast, PEX# for human; Δ pex for yeast)

Line 82 – In seeing the word “similar” it is unclear if the *S. cerevisiae* Pex2 also had deletions in the proteins before purification and whether that affected the inability to detect any ubiquitination by Pex2

Line 69 – rather than say “unknown” it might be prudent to acknowledge previous publications on the role of individual RING peroxins

Referee #3 (Remarks to the Author):

The manuscript by Rapoport and colleagues reports the cryo-EM structure of a membrane-embedded peroxisomal ubiquitin ligase complex (from *T. thermophilus*) that functions to recycle the receptors that mediate the import of proteins into the peroxisomal lumen. Receptor recycling involves the ubiquitination of an N-terminal cysteine residue on the receptor which is the signal for the Pex1-Pex6 ATPase that extracts the receptor from the membrane. In an alternative pathway, poly-ubiquitination of Lys residues at the N-terminus of the receptor targets the receptor to the UPS. The (unnamed) ubiquitin complex comprises three subunits, Pex2, Pex10 and Pex12, each with a RING finger domain. The cryo-EM structure reveals the overall architecture of the complex showing that all subunits share structural homology and therefore a common ancestor. The transmembrane region of each is composed of five helices, and these together define a central trans-membrane pore, whereas the RF domains form the cytosolic segments. Pex10 and Pex12 interact through their RF domains, whereas Pex2 interacts with Pex10 and Pex12 through the transmembrane segments. The structure allows the mapping onto the of disease-associated mutations. Analysis of the RF domains suggested that RF10 is the most canonical, with RF2 also having two zinc-binding sites, with only one zinc site in RF12. Constricting the central trans-membrane by introducing bulky residues, prevented the recycling of receptors and prevented protein import.

Using in vitro ubiquitin ligase assays, the authors show that RF10 is capable of auto-ubiquitination, an activity that is stimulated by RF12. Mutations predicted to disrupt E2 binding to RF10 and ubiquitin to RF12 show reduced (auto)ubiquitination activity.

The authors propose a model whereby a Cys residue within the N-terminus of the receptor is ubiquitinated by RF2 to facilitate receptor extraction by Pex1-Pex6. In the absence of RF2 ubiquitination of the receptor, RF10-RF12 ubiquitinates receptor Lys residues that results in receptor degradation. This is an interesting model, however as discussed below, the authors do not provide strong evidence that RF2 ubiquitinates receptors on Cys residues.

The manuscript is well written, accompanied by generally excellent, clearly labelled figures. The structure determination (cryo-EM of the complex and crystal structure of the RF12 domain) is performed to high quality and resolutions. The quality of the cryo EM structure is supported by reliable FSC curves and good cryo-EM density showing details of amino acid side chain fits. There are many interesting features of the complex, including the novel architecture and visualization of phospholipids and cholesterol, and the explanation of how the RF10/RF12 domains, similar to how some other dimeric RING fingers catalyse ubiquitination with the catalytic RING domain interacting with the E2 and the second RF interacting with ubiquitin.

Questions:

The major questions concern the ubiquitination assays and the assignment of roles of the RF domains of Pex2 and Pex10/Pex12.

1. The basis for assigning Pex2 to the receptor extraction function and Pex10/Pex12 to the receptor UPS pathway wasn't entirely clear. The authors identified potential receptor substrates by using a Pex10 mutant defective in polyubiquitination, which could suggest that Pex10/Pex12 mediate the UPS pathway. However, this mutant would also disrupt the receptor extraction pathway. Was a similar approach applied to Pex2?
2. The ubiquitination assays were performed with the RF domains alone and a test of ubiquitination was auto-ubiquitination. Ideally the authors should test the activity of the whole complex with the authentic receptors. The absence of RF2 activity could be due to the lack of structural context, and/or the fact that the relevant receptor substrates were not used.
3. It isn't clear that the authors' assay would detect substrate ubiquitination on Cys residues. Thiol-esters are relatively unstable. What controls were included to test for this?
4. Evidence that RF2/Pex2 functions as an E3 ligase would greatly strengthen this manuscript. The authors note that they have been unable to demonstrate that in vitro (lines 214-215), but they do not indicate what has been tested. In addition to testing Pex2 activity in the context of the whole complex (in which case RF10/RF12 would need to be inactivated), and with the relevant substrates, the authors should at least attempt to show an interaction between RF2 and a relevant E2.
5. To test that Pex2 is the subunit responsible for receptor extraction, the authors could use the in vivo assay (lines 205-230), by combining the Pex5 C6A mutant with the RF2 (P223A/R224D/D257A) mutant. If RF2 mediates receptor extraction, the combination of the Pex5 C6A and RF2 mutants should give the same phenotype as mutating Pex5 and RF2 alone.
6. The authors propose that RF2-mediated ubiquitination determines receptor extraction, whereas RF10/RF12-mediated ubiquitination determines receptor degradation. What determines or regulates which pathway is active?
7. Did the authors test the consequence of disease-associated mutations on E3 ligase activity?

Author Rebuttals to Initial Comments:

Referee #1 (Remarks to the Author):

This is an elegant structure-function study of a novel membrane-associated E3 ligase Pex2/10/12 of the yeast peroxisome. The cryo-EM structure was determined to an average resolution of 3.1 Å, with the help of a protein surface PEGylation which likely improved the particle orientation distribution in the EM grids as well as the production of a Fab fragment that increased the size of the small complex (150 kDa) and facilitated particle image alignment. The cryo-EM analysis is well performed. The only weakness is the relatively low resolution of the largely flexible RF2 domain. But the authors have remediated the problem by determining a high-resolution crystal structure of the truncated domain. The structure is unexpected and striking, revealing a constitutive open channel of 10 Å formed by 15 transmembrane helices, five from each subunit. Mutagenesis and functional assays demonstrate that this aqueous channel is utilized by the N-termini of the recycling receptors from the peroxisome lumen to reach the catalytic RF domains on the cytosolic side of the peroxisome membrane for ubiquitination. Another key insight is the identification of the mono-ubiquitination and poly-ubiquitination mechanisms in Pex2/10/12 ligase complex. The manuscript is very well written, and the Figures are well prepared and illustrative. This work represents an important advance on peroxisome biogenesis that will be of interest to the wide audience of Nature.

Minor corrections

1. There is an unintended horizontal white line across the yellow histogram bars in Fig. 4e.

The error was corrected.

2. Page 7 line 149: The sentence does not sound logic and may need attention.

The sentence was shortened and should make sense now.

Referee #2 (Remarks to the Author):

This interesting paper by Feng et al. is a potentially exciting advance because the three RING-domain peroxins, Pex2, Pex10 and Pex12, residing in the peroxisomal membrane serve as E3 ligases for several aspects of peroxisome dynamics, such as peroxisome biogenesis, PTS receptor quality control (RADAR) and pexophagy. Structural information provided by this paper is therefore potentially interesting, particularly as it relates to the roles of these RING peroxins in PTS receptor mono- and poly-ubiquitination reactions. In this paper, the authors focus on the structure of the hetero-trimeric complex of Pex2, Pex10 and Pex12, primarily of fungal origin. Although several PTS

receptors (Pex5, Pex7) and their co-receptors (Pex18/20/21) are ubiquitinated by this RING subcomplex, the authors have studied only Pex5 ubiquitination physiologically and present models for how the RING subcomplex could be involved in mono- and poly-ubiquitination of targets during the peroxisomal matrix protein import cycle.

The strength of the paper is that it is structurally excellent with the determination of RING complex structures from several sources contributing to the analysis. It is also original. The data are of high quality, although the description of the methods is sub-optimal to engender reproducibility. Although the topic is of potential interest and the structural work is sound, several issues, including the interpretation of the data and their physiological relevance suggest that it is too preliminary at present and much further work would be needed.

1. The authors could not detect any mono-ubiquitination by the RING subcomplex, which normally uses the endogenous Pex4 as the E2 enzyme. Instead the authors used Ubc4 as the E2 enzyme for all their in vitro studies. Thus, a key aspect of the role of this RING subcomplex in PTS receptor recycling has not been studied or recapitulated. Although the authors present a model for how Pex2 might be involved in Pex5 mono-ubiquitination (lines 140-147 and Fig. 4h), it is speculative, without data to back it up, and contradicts previous publications suggesting that Pex12 facilitates the Pex4-dependent mono-ubiquitination of Pex5 (Platta et al., 2009; PMID: 19687296).

We have included additional data that provide evidence that Pex2 is involved in monoubiquitination (Extended Data Fig 10d). We overexpressed a Pex5 mutant that lacks the two lysines for polyubiquitination in a yeast strain that carries a mutation in RING finger 2 (RF2), which should compromise the interaction with the E2 enzyme. Although the Pex5 mutant still contains the key cysteine for monoubiquitination, protein import into peroxisomes was reduced by ~50% compared to that with wild type Pex5 (Extended Data Fig 10d). These data support our model that RF2 catalyzes monoubiquitination. The interaction with the E2 enzyme is not totally abolished, consistent with our other data (Fig. 4f). We added new text to discuss these data (lines 227-235).

With regard to RF12, our structure shows that this RING finger lacks the features required for binding an E2 enzyme and is therefore unable to mediate ubiquitination.

2. The authors have studied Pex5 poly-ubiquitination (which is involved in the RADAR pathway) and present a nice model suggesting that Pex10, stimulated by Pex12, does this poly-ubiquitination. However, this too contradicts a previous report suggesting that Pex2 mediates the Ubc4-dependent polyubiquitination (Platta et al., 2009; PMID: 19687296), although it does agree with another report showing that Pex10 is the E3 ligase involved in Ubc4-dependent poly-ubiquitination of Pex5 (Williams C et al 2008; PMID: 18644345).

We cited the paper by Williams et al. (2008), which is much more convincing than the one by Platta et al (2009). Williams et al. showed that RF10 is only active with Ubch5a (a homolog of yeast Ubc4),

but inactive with Pex4. These results are in excellent agreement with ours.

3. It is unclear from the manuscript what the site/s of poly-ubiquitination are in Pex5, which is important because the mono-ubiquitination of Pex5 is at a Cys near the N-terminus, but the poly-ubiquitination is at Lys residues elsewhere (Lys18 and Lys24 in yeast Pex5) and the physiological consequences of modifications at these sites are very different.

As we pointed out in the manuscript (line 224), previous work by other groups (refs. 29 and 30) have clearly established that Lys18 and Lys24 of Pex5 are the only two residues that are polyubiquitinated. The data shown in Fig. 4g support this conclusion.

4. The authors do not present their data carefully – one example is that many sites of mutation in human PEX10 (and other proteins) are presented incorrectly in the text and one Figure. As examples, the reference to H310D/Q is incorrectly cited in hPEX10, as is L297 and C327 (Fig. 7b). Extended Data Fig. 7b has several incorrectly labeled residues in hPEX10 and likely TtPEX10. The authors should check all the numbering of residues from all species. In Fig. 4d, should C353S be C354S?

We apologize for the mistakes. There are two isoforms of human PEX10 and we used the long isoform for numbering. We now mention this in the legend to Extended Data Fig. 7. We also corrected the mistakes in Fig. 4d and Extended Data Fig. 7b.

5. An unexpected surprise is that this paper claims that all the RING peroxins have 5 transmembrane domains (TMDs). Given that there are contradictory reports in the literature suggesting otherwise - e.g. Pex12 is reported to have either an even (N and C- termini of PEX12 face the cytosol, Okumoto et al., 1998; PMID: 9632816), or odd (Albertini et al., 2001; PMID: 11370741) number of TMDs - it is obligatory that the authors provide biochemical evidence for the membrane topology and orientation of the termini, minimally for at least one of the RING peroxins.

Previous data in the literature on the topology of Pex2, Pex10, and Pex12 were indeed conflicting, in large part because hydropathy predictions were incorrect and the methods for topology determination (antibody accessibility of tags and protease protection) were inadequate. The existence of 5 TMs in each of the three proteins is therefore indeed surprising. However, our structure firmly establishes the topology of the three Pex proteins. The density map allows us to unambiguously trace the trans-membrane segments of all three proteins, as confirmed by reviewer #1 (a cryo-EM specialist). There is not the slightest doubt that the N-termini of all proteins are located in the lumen and the C-termini, including the RFs, in the cytosol. The cytosolic localization of the RFs is, of course, required because of the localization of the E1 and E2 enzymes and of ubiquitin. It should also be noted that our structure is in agreement with AlphaFold predicting 5 TMs of the correct orientation for the three proteins from all species. Given that structure determination is arguably the best way to determine the topology of a membrane protein, we feel strongly that there is no need for further experiments on the topologies.

The paper by Okumoto et al. (1998) claimed that the N-terminus of Pex12 is located in the cytosol on the basis of accessibility of an N-terminal tag to antibodies. The authors overexpressed only this subunit of the ligase complex, so it was in excess over the other subunits and might have aggregated or misfolded, they did not ascertain that the observed puncta correspond to peroxisomes, and they did not exclude non-specific binding of the FLAG antibodies. These problems could be the reason for why the authors reached the wrong conclusion.

6. The authors should cite that PEX2 is a E3 ligase for mammalian pexophagy (Sargent et al., 2016; PMID: 27597759; because this points out that there are other targets, such as PMP70, and a different site, Lys209, on PEX5 for mono-ubiquitination).

It is possible that Pex2 has targets in addition to the import receptors. Because our paper does not deal with pexophagy, we feel that there is no need to discuss this point.

7. Open pore discussion starting line 127 – In peroxisomes, it is proposed that Pex8 is part of the importomer (Agne et al., 2003; PMID: 12667447) and that it may be involved in cargo release (Ma et al., 2013; PMID: 23902771). The authors should use caution to call it an “open pore” before ruling out that it might be occluded in vivo prior to cargo release.

We indeed cannot exclude that the pore is closed under certain conditions and have therefore phrased the text more cautiously (line 127 and lines 270/271). However, Pex8 cannot generally occlude the pore, simply because it is only found in fungi. In addition, AlphaFold predicts with high confidence that Pex8 consists entirely of HEAT-repeats and does not contain unstructured termini similar to all import receptors. It is therefore unlikely that Pex8 inserts a segment into the pore and plugs it.

8. In the model in Fig. 4h, where does cargo release fit in? Without this step, what would prevent Pex5 bound PTS cargo from being exported? Also, the statement about the pore being constitutively “open” is speculation at this stage (lines 262,263 – see previous comment regarding Pex8).

As mentioned under point #7, we have phrased the text more cautiously. Our paper does not address the mechanism of cargo release, but the pore of the ligase complex is too narrow to allow the cargo to exit through it.

Minor

The authors use multiple names and spellings for *T. thermophilus* (including *myceliophthra* in Extended Data Fig. 1)

We have corrected the inconsistencies.

There is inconsistent use of yeast and mammalian nomenclature for peroxins (Pex# for yeast, PEX# for human; Dpex for yeast)

We have corrected the inconsistencies.

Line 82 – In seeing the word “similar” it is unclear if the *S. cerevisiae* Pex2 also had deletions in the proteins before purification and whether that affected the inability to detect any ubiquitination by Pex2

The constructs of the *S. cerevisiae* ligase complex are all full-length proteins without any truncation. This is now mentioned in the legend to Extended Data Fig. 2.

Line 69 – rather than say “unknown” it might be prudent to acknowledge previous publications on the role of individual RING peroxins

As requested, we have changed the sentence and cite two papers (lines 68/69). As the reviewer pointed out above, the data in the literature on the individual RING peroxins are contradictory.

Referee #3 (Remarks to the Author):

The manuscript by Rapoport and colleagues reports the cryo-EM structure of a membrane-embedded peroxisomal ubiquitin ligase complex (from *T. thermophilus*) that functions to recycle the receptors that mediate the import of proteins into the peroxisomal lumen. Receptor recycling involves the ubiquitination of an N-terminal cysteine residue on the receptor which is the signal for the Pex1-Pex6 ATPase that extracts the receptor from the membrane. In an alternative pathway, poly-ubiquitination of Lys residues at the N-terminus of the receptor targets the receptor to the UPS. The (unnamed) ubiquitin complex comprises three subunits, Pex2, Pex10 and Pex12, each with a RING finger domain. The cryo-EM structure reveals the overall architecture of the complex showing that all subunits share structural homology and therefore a common ancestor. The transmembrane region of each is composed of five helices, and these together define a central trans-membrane pore, whereas the RF domains form the cytosolic segments. Pex10 and Pex12 interact through their RF domains, whereas Pex2 interacts with

Pex10 and Pex12 through the transmembrane segments. The structure allows the mapping onto the of disease-associated mutations. Analysis of the RF domains suggested that RF10 is the most canonical, with RF2 also having two zinc-binding sites, with only one zinc site in RF12. Constricting the central trans-membrane by introducing bulky residues, prevented the recycling of receptors and prevented protein import.

Using in vitro ubiquitin ligase assays, the authors show that RF10 is capable of auto-ubiquitination, an activity that is stimulated by RF12. Mutations predicted to disrupt E2 binding to RF10 and ubiquitin to RF12 show reduced (auto)ubiquitination activity.

The authors propose a model whereby a Cys residue within the N-terminus of the receptor is ubiquitinated by RF2 to facilitate receptor extraction by Pex1-Pex6. In the absence of RF2 ubiquitination of the receptor, RF10-RF12 ubiquitinates receptor Lys residues that results in receptor degradation. This is an interesting model, however as discussed below, the authors do not provide strong evidence that RF2 ubiquitinates receptors on Cys residues.

The manuscript is well written, accompanied by generally excellent, clearly labelled figures. The structure determination (cryo-EM of the complex and crystal structure of the RF12 domain) is performed to high quality and resolutions. The quality of the cryo EM structure is supported by reliable FSC curves and good cryo-EM density showing details of amino acid side chain fits. There are many interesting features of the complex, including the novel architecture and visualization of phospholipids and cholesterol, and the explanation of how the RF10/RF12 domains, similar to how some other dimeric RING fingers catalyse ubiquitination with the catalytic RING domain interacting with the E2 and the second RF interacting with ubiquitin.

Questions:

The major questions concern the ubiquitination assays and the assignment of roles of the RF domains of Pex2 and Pex10/Pex12.

1. The basis for assigning Pex2 to the receptor extraction function and Pex10/Pex12 to the receptor UPS pathway wasn't entirely clear. The authors identified potential receptor substrates by using a Pex10 mutant defective in polyubiquitination, which could suggest that Pex10/Pex12 mediate the UPS pathway. However, this mutant would also disrupt the receptor extraction pathway. Was a similar approach applied to Pex2?

We show that mutations in RF10 and RF12 that compromise polyubiquitination in vitro have no protein import defect in vivo (Fig. 4a-d). Specifically, a mutation in Pex10 (R324A) that abolishes the interaction with the E2 enzyme, does not reduce import. Further evidence that polyubiquitination is not required for import comes from the data in Fig. 4g, where we show that the targets of polyubiquitination, Lys18 and Lys24 of Pex5, are not required. We used a similar approach for Pex2 (Fig. 4e) and showed that mutations predicted to affect the interaction with the E2 enzyme reduce protein import when combined with Pex10 mutants defective in polyubiquitination. We have now added additional data that support the idea that Pex2 catalyzes monoubiquitination, as the

combination of the Pex2 mutations and Lys18/24 mutations in Pex5 has a significant import defect (Extended Data Fig 10d).

2. The ubiquitination assays were performed with the RF domains alone and a test of ubiquitination was auto-ubiquitination. Ideally the authors should test the activity of the whole complex with the authentic receptors. The absence of RF2 activity could be due to the lack of structural context, and/or the fact that the relevant receptor substrates were not used.

We tried to recapitulate monoubiquitination with the full-length ligase complex, using either wild-type or mutant Pex10 and the E2 enzyme complex containing Pex4 and the soluble domain of Pex22 (Pex4/Pex22s). The ligase complex was tested for poly- and mono-ubiquitination of purified Pex5. Although we could readily detect the formation of a thioester of Pex4, we were unable to detect monoubiquitinated Pex5 or even autoubiquitination of the ligase ***Redacted***. We also failed to detect an interaction between the ligase and ubiquitin-conjugated E2 (Ub~Pex4/Pex22s), regardless of whether we pulled on the ligase or the E2 enzyme ***Redacted***.

We believe that monoubiquitination may require the insertion of the N-terminus of Pex5 into the ligase pore, which may be impossible to achieve by simply adding the components together. It is also conceivable that an unknown component is missing. We added a few words to mention these possibilities (lines 235-238) Obviously, recapitulating monoubiquitination remains an important goal for the future, but is beyond the scope of the present paper.

3. It isn't clear that the authors' assay would detect substrate ubiquitination on Cys residues. Thiol-esters are relatively unstable. What controls were included to test for this?

Since we can see the ubiquitin-conjugated E2 enzyme in the in vitro assay ***Redacted***, the stability of the thiol-ester is not the reason for our inability to detect monoubiquitination.

4. Evidence that RF2/Pex2 functions as an E3 ligase would greatly strengthen this manuscript. The authors note that they have been unable to demonstrate that in vitro (lines 214-215), but they do not indicate what has been tested. In addition to testing Pex2 activity in the context of the whole complex (in which case RF10/RF12 would need to be inactivated), and with the relevant substrates, the authors should at least attempt to show an interaction between RF2 and a relevant E2.

We now provide additional evidence that Pex2 mediates monoubiquitination (see answer to points #1 and #5).

5. To test that Pex2 is the subunit responsible for receptor extraction, the authors could use the in vivo assay (lines 205-230), by combining the Pex5 C6A mutant with the RF2 (P223A/R224D/D257A) mutant. If RF2 mediates receptor extraction, the combination of the Pex5 C6A and RF2 mutants should give the same phenotype as mutating Pex5 and RF2 alone.

Expressing the Pex5 C6A mutant instead of wild-type Pex5 abolishes all peroxisomal protein import (shown in multiple papers), so combining this mutant with a mutation in Pex2 would have no additional effect. However, experiments similar to the one suggested by the reviewer can be done by overexpressing Pex5 mutants in wild-type or Pex2 mutant cells containing endogenous Pex5. In the previously submitted manuscript, we already showed that overexpression of the C6A mutant in wild-type cells causes only a slight protein import defect, as the mutant protein can be extracted from the ligase pore by the polyubiquitination/degradation pathway, so that it does not interfere with the endogenous Pex5 protein. We now provide new data in which we overexpressed Pex5 mutants in a strain that contains a RF2 mutation (L1) in a loop that likely affects the interaction with the E2 enzyme (new Extended Data Fig. 10d). As expected, expression of the C6A mutant causes only a slight import defect, again because the polyubiquitination pathway can clear the pore for endogenous Pex5. However, expression of the polyubiquitination-defective K18/K24 mutant reduces import by ~50%, supporting our claim that RF2 catalyzes monoubiquitination. The interaction of RF2 with the E2 enzyme is not totally abolished, consistent with our other data (Fig. 4f).

6. The authors propose that RF2-mediated ubiquitination determines receptor extraction, whereas RF10/RF12-mediated ubiquitination determines receptor degradation. What determines or regulates which pathway is active?

Normal receptor recycling/extraction is probably fast so that polyubiquitination does not play a significant role. When receptor recycling is compromised and the dwell time of the receptor in the pore is increased, the receptor can move sideways and be polyubiquitinated. In yeast, lateral movement requires the displacement of the plug. It remains unclear under which physiological conditions receptor recycling is compromised. We now mention this point (lines 253/254).

7. Did the authors test the consequence of disease-associated mutations on E3 ligase activity?

We tested one disease mutant in human PEX10 (H310D), which corresponds to Pex10 (H303D) in yeast both in vivo and in vitro (Fig. 4b and c). Another tested mutation, Pex10R324A, is similar to a reported disease mutation (human PEX10R331Q).

Redacted*

Reviewer Reports on the First Revision:

Referees' comments:

Referee #2 (Remarks to the Author):

The revision addresses all of my major concerns but leaves these minor ones.

1. thermophilus spelling errors remain uncorrected in the Extended Data section
2. It would help the reader to know which isoform is used in the numbering of human Pex10 in Extended Data Fig. 1. Without this the different numbering system used in Extended data Fig. 7 is still confusing.

Referee #3 (Remarks to the Author):

Following the requests of reviewers 2 and 3, the authors have added new data that supports their model that Pex2 mediates Pex5 import through Cys6 mono-ubiquitination (Extended Data Fig. 10d).

However, the authors have been unable to show (i) that either Pex2/RF2 or the holocomplex catalyses monoubiquitination of Pex5 *in vitro* (or *in vivo*) and, related (ii) that Pex2/RF2 possesses any E3 ligase activity and (iii) that Pex2/RF2 interacts with an E2 enzyme. This absence of direct evidence for the assigned role of Pex2/RF2 reduces the impact of this study.

If publication of this study proceeds, it should clearly state these limitations. The 'yet' should be removed from the sentence on line 235.

Author Rebuttals to First Revision:

Point-by-point response to the remaining issues raised by the reviewers (our response is in red):

Referee #2 (Remarks to the Author):

The revision addresses all of my major concerns but leaves these minor ones.

1. thermophilus spelling errors remain uncorrected in the Extended Data section

The spelling errors were corrected.

2. It would help the reader to know which isoform is used in the numbering of human Pex10 in Extended Data Fig. 1. Without this the different numbering system used in Extended data Fig. 7 is still confusing.

We now mention in the legend to Extended Data Fig. 1 that the long isoform of human PEX10 (UniProtKB – O60683-2) was used for sequence alignment.

Referee #3 (Remarks to the Author):

Following the requests of reviewers 2 and 3, the authors have added new data that supports their model that Pex2 mediates Pex5 import through Cys6 mono-ubiquitination (Extended Data Fig. 10d).

However, the authors have been unable to show (i) that either Pex2/RF2 or the holocomplex catalyses monoubiquitination of Pex5 in vitro (or in vivo) and, related (ii) that Pex2/RF2 possesses any E3 ligase activity and (iii) that Pex2/RF2 interacts with an E2 enzyme. This absence of direct evidence for the assigned role of Pex2/RF2 reduces the impact of this study.

If publication of this study proceeds, it should clearly state these limitations. The 'yet' should be removed from the sentence on line 235.

We clearly state the limitations of our study, i.e. that we were unable to demonstrate mono-ubiquitination by RF2 or Pex2 in vitro (lines 222-225). The "yet" in the sentence was deleted, as requested by the reviewer. We appreciate that the reviewer agrees with us that our new data, added to the revised manuscript, support our conclusion "that Pex2 mediates Pex5 import through Cys6 mono-ubiquitination (Extended Data Fig. 10d)